# Spatiotemporal Variation, Sources, and Secondary Transformation Potential of VOCs in Xi'an, China

Mengdi Song[1], Xin Li[1,*], Suding Yang[1], Xuena Yu[1], Songxiu Zhou[1], Yiming Yang[1], Shiyi Chen[1], Huabin Dong[1], Keren Liao[1], Qi Chen[1], Keding Lu[1], Ningning Zhang[2], Junji Cao[2], Limin Zeng[1], Yuanhang Zhang[1]

[1]State Key Joint Laboratory of Environmental Simulation and Pollution Control, College of Environmental Sciences and Engineering, Peking University, Beijing 100871, China
[2]Key laboratory of Aerosol Chemistry and Physics, SKLLQG, Institute of Earth Environment, Chinese Academy of Science, Xi'an 710061, China

*Correspondence to*: Xin Li (li_xin@pku.edu.cn)

**Abstract.** As critical precursors of ozone ($O_3$) and secondary organic aerosols, volatile organic compounds (VOCs) play a vital role in air quality, human health, and climate change. In this study, a campaign of comprehensive field observations and VOC grid sampling was conducted in Xi'an, China from June 20 to July 20, 2019 to identify the spatiotemporal concentration levels, sources, and secondary transformation potential of VOCs. During the observation period, the average VOC concentrations at the Chanba (CB), Di Huan Suo (DHS), Qinling (QL), and gridded sampling sites were 27.8 ± 8.9, 33.8 ± 10.5, 15.5±5.8, and 29.1±8.4 ppb, respectively. Vehicle exhaust was the primary source of VOC emissions in Xi'an, and the contributions of vehicle exhaust to VOCs at the CB, DHS, and QL sites were 41.3%, 30.6%, and 23.6–41.4%, respectively. While industrial emissions were the second-largest source of VOCs in urban areas, contributions from ageing sources were high in rural areas. High potential source contribution function values primarily appeared in eastern and southern Xi'an near the sampling site, which indicates that Xi'an exhibits a strong local VOC source. Moreover, alkenes, aromatics, and oxygenated VOCs played a dominant role in secondary transformation, which is a major concern in reducing $O_3$ pollution in Xi'an.

## 1 Introduction

Atmospheric pollution in China is characterized by frequent secondary pollution, which is primarily reflected by the yearly increasing ozone ($O_3$) concentrations and proportion of secondary organic components (SOA) in $PM_{2.5}$ (Lu et al., 2018; Huang et al., 2014). The increasingly severe secondary pollution has restricted improvements in Chinese ambient air quality. Volatile organic compounds (VOCs) are vital precursors of $O_3$ and SOA, and reducing VOC emissions is crucial for controlling $O_3$ and $PM_{2.5}$ pollution (Jin and Holloway, 2015; Yuan et al., 2013).

VOC sources can be divided into biogenic and anthropogenic sources. Although the VOC emissions from biological sources are more than 10 times that of anthropogenic sources globally (Roger and Janet, 2003), anthropogenic sources often dominate the urban atmospheric environment and exert a substantial influence on the formation of $O_3$ and SOA in the atmosphere (Ahmad et al., 2017; Cuisset et al., 2016; Schwantes et al., 2016). Anthropogenic VOC sources primarily include vehicle

exhaust, industrial, paint solvent usage, combustion, and fuel evaporation sources. In recent years, the source apportionment of observation results found that the contribution of vehicle exhaust sources to VOCs plays a leading role in VOC emissions in China, reaching 22–58% (Jia et al., 2016; Liu et al., 2016a; Liu et al., 2020; Zhang et al., 2019). However, the VOC sources exhibit clear differences in various regions. For example, paint solvent usage is the main source of VOCs in Langfang and

Xiamen City (Zhang et al., 2019; Zhuang et al., 2019), while the primary sources of VOCs in Wuhan City are combustion and vehicle exhaust sources (Shen et al., 2020). The results of VOC emission inventory research indicate that the contribution of vehicle exhaust sources to VOC emissions is gradually decreasing, and industrial sources and paint solvent usage have become the primary sources of VOCs in China (Li et al., 2019b; Wu et al., 2016b).

$O_3$ has become one of the most important pollutants affecting the air quality of the Fenwei Plain. According to the 2018 and

2019 Report on the State of the Ecology and Environment in China released by the Ministry of Ecology and Environment the People's Republic of China (http://www.mee.gov.cn/), the annual average evaluation value of $O_3$ concentration in the Fenwei Plain exceeded the Chinese annual evaluation standard (160 μg/m$^3$) for at least two consecutive years. Moreover, the average 90[th] percentile $O_3$ daily maximum eight-hour concentration in the Fenwei Plain was 171 μg/m$^3$ (298.15 K, 1013.25 hPa), which was 4.2% higher than that in 2018. Xi'an, one of the most famous historic and megacities in Fenwei Plain with a quickly

growing economy, has also experienced severe $O_3$ pollution in recent years. Based on the Report on the State of the Environment in Xi'an (http://xaepb.xa.gov.cn/), the average 90[th] percentile $O_3$ daily maximum eight-hour concentration in Xi'an exceeded the national standards, reaching 169.5 μg/m$^3$, 164.9 μg/m$^3$, and 166 μg/m$^3$ (298.15 K, 1013.25 hPa) in 2017, 2018, and 2019, respectively. In addition, according to statistics released by the Ministry of Public Security, as of the end of 2018, the number of motor vehicles in Xi'an reached 3 million, making Xi'an one of the top eight cities in China in terms of

number of motor vehicles. Considering the increasing $O_3$ concentration and high amount of pollution in Xi'an, studies are imperative in exploring the characteristics, sources, and precursors (VOCs and NOx) of $O_3$ to support the design and implementation of pollution control strategies in Xi'an.

In recent years, numerous VOC observations have been made in many regions of China, particularly in the Beijing-Tianjin-Hebei, Yangtze River Delta, Pearl River Delta, and Chengdu-Chongqing regions. However, there are few studies on Xi'an.

Therefore, our understanding of the spatiotemporal characteristics, source contributions, and secondary transformation potential of VOCs in Xi'an remains limited. To elucidate the concentration levels, source characteristics, and secondary conversion ability of $O_3$ precursors, we conducted comprehensive field observations and VOC grid sampling in Xi'an from June 20 to July 20, 2019. The primary objectives of this study are to (1) determine the temporal and spatial characteristics of atmospheric VOCs in Xi'an, (2) illuminate the characteristics of VOCs during $O_3$ pollution events, (3) identify critical

precursors in Xi'an, and (4) explore VOC sources in Xi'an.

## 2 Observation and methods

### 2.1 Observation campaign description

To obtain the temporal and spatial distributions of summertime VOCs in Xi'an, a field observation and VOC grid sampling campaign was applied in this study. The field observation campaign occurred from June 20 to July 20, 2019. Based on an in-person investigation, the prevailing winds, and the electricity supply, three sites were chosen for the field observation campaign (Figure 1): the Chanba (CB; 34°20'12"N, 109°01'35"E), Di Huan Suo (DHS; 34°13'49"N, 108°52'58"E), and Qinling (QL; 34°04'11"N, 108°20'31"E) sites, which were located on upwind, downtown, and downwind areas of Xi'an City, respectively. The CB site was at the national environmental air quality monitoring urban station (approximately 10 m above ground level) in the Baqiao District, which was close to the third ring road, one of the main traffic thoroughfares in Xi'an. The DHS site was located on the roof of a four-story building (approximately 15 m above ground level) at the Institute of Earth Environment, Chinese Academy of Science in the Yanta District, which was close to the second ring road. The QL site was located in Zhouzhi County, which was adjacent to the Qinling National Botanical Garden. The VOC measurement instrument used in the field observation campaign was an online gas chromatography system that was equipped with a mass spectrometer and flame ionization detector (GC–MS/FID). The GC–MS/FID instrument utilized a dual gas path separation method. The sample gas, after water and $CO_2$ were removed, captured VOC components through an ultra-low temperature system (-160 °C), and the gas chromatography analysis system was utilized after thermal desorption. The oven temperature was programmed at 37 °C maintained for 5 min initially, then raised to 120 °C at 5 °C min$^{-1}$ holding for 5 min and later to 180 °C at 6 °C min$^{-1}$ holding for 5 min. The low carbon number ($C_2$–$C_5$) compounds were separated on an $Al_2O_3$/KCl PLOT column and quantified using the FID (200 °C). The high carbon number ($C_5$–$C_{10}$) compounds were separated on a DB-624 column and quantified using MS (230 °C). External and internal standard gas produced by The Linde Group in the United States were used to calibrate the GC–MS/FID. External standard gas were used to calibrate the GC–MS/FID weekly during the campaign to ensure quantitative accuracy. In addition, the instruments were also daily calibrated by internal standard gases (Bromochloromethane, 1,4-Dichlorobenzene, Chlorobenzene, and Fluorobromobenzene) to ensure the stability of the instrument. During the observation period, the GC–MS/FID was in sufficient working condition, and the square correlation coefficients of the VOCs work curves were greater than 0.99. The method detection limits (MDLs) for each VOC compound were calculated according to the TO-15 standard of the United States Environmental Protection Agency (EPA) (Liu et al., 2008b), and the MDL for all measured VOC compounds in this study ranged from 0.002 to 0.121 ppb. $O_3$, NO, $NO_2$, and NOx concentrations were continuously monitored using i-series chemiluminescence instruments.

VOC gridded samples were collected at each site for two days (July 1 and July 14, 2019) and twice a day at 7:00 China Standard Time (CST) and 15:00 (CST). The gridded sampling site were chosen based on the technical regulations for selecting ambient air quality monitoring stations (HJ 664-2013) and method for selection of Photochemical Assessment Monitoring Stations (EPA). According to the prevailing wind direction and to ensure coverage of all urban areas in Xi'an, 20 gridded sampling sites were selected for this study (Figure 1). Detailed sampling site information is shown in Table 1. A total of 80

ambient air samples with a frequency of 4 samples per site were collected and each sample was stored in a 3.2-L SiloniteTM

canister (Entech Instrument, United States). Before VOC gridded sampling, the Silonite$^{TM}$ canisters were cleaned with high purity nitrogen using the Entech 3100 canister cleaning system, and then they were evacuated to a vacuum. Instantaneous sampling method was adopted for ambient air sample collection with a sampling duration of approximately 2 min. VOCs in the sampled air were analyzed using a GC–MS/FID system, which was the same as that used for online measurements, but it was running in off-line mode. In this study, blank sample tests were performed on the VOC gridded sampling in each sampling

period (July 1 7:00, July 1 15:00, July 14 7:00, and July 14 15:00). The concentration of all VOCs species in the blank sample is below the MDLs, indicating that the canisters were not contaminated during transportation.

## 2.2 Photochemical reactive activity parameterization

The loss rates of VOCs that react with OH radicals ($L_{OH}$) and the $O_3$ formation potential (OFP) can be used to characterize VOC photochemical activity (Carter, 2010; Niu et al., 2016; Wu et al., 2016a). $L_{OH}$ and OFP can be calculated using Eqs. (1)

and (2), respectively.

$$L_{OH} = \sum_{i}^{n} k_{OH_i} \times [VOC\ (molecule \cdot cm^{-3})]_i \tag{1}$$

$$OFP = \sum_{i}^{n} MIR_i \times [VOC\ (ppb)]_i \tag{2}$$

where n represents the number of VOCs, $[VOC]_i$ represents the $i^{th}$ VOC species concentration, $k_{OHi}$ represents the rate coefficient for the reaction of the $i^{th}$ VOC species with OH radical ($molecule^{-1} \cdot cm^3 \cdot s^{-1}$), and $MIR_i$ is the maximum incremental

reactivity for the $i^{th}$ VOC species. The $k_{OHi}$ and MIR for each VOC specie were taken from the updated Carter research results (http://www.engr.ucr.edu/~carter/reactdat.htm).

Specific VOC ratios are often used to calculate the air mass photochemical age or OH exposure (Jimenez et al., 2009; Roberts et al., 1984). OH exposure can be calculated using Eq. (3).

$$OH\ exposure = \frac{1}{(k_{voc1} - k_{voc2})} \times \left[ \ln \frac{[voc1]}{[voc2]} \Big|_{t=0} - \ln \frac{[voc1]}{[voc2]} \right] \tag{3}$$

where OH exposure represents OH radical concentration multiplied by the reaction time ($\Delta t$); $k_{voc1}$ and $k_{voc2}$ represent rate coefficients for the reaction of the VOC species with OH radical ($molecule^{-1} \cdot cm^3 \cdot s^{-1}$); $\frac{[VOC1]}{[VOC2]} \Big|_{t=0}$ represents the initial emission ratio of specific VOCs, which can be replaced by the highest concentration ratio in periods where the photochemical reaction is weak; and $\frac{[VOC1]}{[VOC2]}$ represents the concentration ratio of the specific VOCs in the atmosphere.

## 2.3 Positive matrix factorization (PMF)

The PMF analysis model was first proposed by Paatero and Tapper (1994). For more than two decades, PMF has been widely used to identify and quantify major sources of VOCs (He et al., 2019; Li et al., 2015; Miller et al., 2002; Pallavi et al., 2019; Song et al., 2007; Yuan et al., 2010). The definition and usage of the PMF model is described elsewhere in detail (Liu et al., 2020; Song et al., 2018; Su et al., 2019), and only a brief description is provided here.

In this study, the PMF 5.0 model (EPA) was used to analyze the VOC sources at the CB, DHS, and QL sites. VOCs tracers were selected according to the reported typical emission source profiles in China (He et al., 2015; Liu et al., 2008a; Song et al., 2018). Only those tracers which have data coverage greater than 75% during the campaign and have 65% measured concentrations above the MDL were included in the PMF analysis (c.f. Liu et al., 2020). In this study, the number of VOC tracers (input species) at the CB, DHS, and QL sites were 32, 34, and 32, respectively. A concentration (Conc.) file and uncertainty (Unc) file are required by PMF. For the concentration file, data below the detection limit were assigned with MDL/2, and the missing data were substituted with median concentration. The uncertainty is calculated using Eq. (4) as follows (Brown et al, 2015; Liu et al, 2016a):

$$\text{Unc} = \begin{cases} \sqrt{(\text{Error Fraction} \times \text{Conc.})^2 + (0.5 \times \text{MDL})^2} & \text{Conc.} > \text{MDL} \\ \frac{5}{6} \times \text{Conc.} & \text{Conc.} \le \text{MDL} \\ 4 \times \text{median conc.} & \text{Missing data} \end{cases} \qquad (4)$$

In this study, the PMF factor numbers were explored from 4 to 8 for the optimal solution in the three sites. $Q_{true} / Q_{robust}$ and $Q_{true} / Q_{expected}$ are two important parameters for characterizing the rationality of the PMF results (Brown et al., 2015). After comparing the PMF results, $Q_{true} / Q_{robust}$ ratio, and $Q_{true} / Q_{expected}$ ratio, a seven-factor PMF solution was selected for VOC source apportionments in the three field observation sites. The $Q_{true} / Q_{robust}$ values at the CB, DHS, and QL sites were all 1.0. The $Q_{true} / Q_{expected}$ values at the CB, DHS, and QL sites were 1.3, 1.2, and 1.0, respectively. In addition, an Fpeak parameter, from -1.0 to 1.0 (step of 0.1), was used to rotate the PMF factors for a superior solution (Sun et al., 2012). In this study, the factor rotating results were not significantly different than the non-rotation results. Thus, the results used in this study were from the runs with zero-Fpeak.

**2.4 Conditional probability function (CPF) analyses**

A CPF analysis can be used to explore the PMF-identified VOC source impacts at varying wind directions and speeds (Huang and Hsieh, 2019; Liu et al., 2016a). The CPF can be calculated using Eq. (5) as follows (Ashbaugh et al., 1985; Uria-Tellaetxe and Carslaw, 2014):

$$\text{CPF} = \frac{m_{\theta,\mu,75per}}{n_{\theta,\mu}} \qquad (5)$$

where $m_{\theta,\mu,75per}$ is the number of samples in the wind direction θ and wind speed interval μ with a VOC concentration greater than the 75[th] percentile concentration, and $n_{\theta,\mu}$ is the total number of samples in the same wind direction and speed interval.

**2.5 Cluster and potential source contribution function (PSCF) analysis**

Cluster analyses of backward trajectories are widely used to determine a dominant air mass direction and potential origin direction of pollutants at the study sites (Hong et al., 2019; Liu et al., 2016a; Liu et al., 2019). In this study, the 24-h backward trajectories (1-h intervals) of air masses arriving 100 m above ground level were calculated using the MeteoInfoMap software. Air mass reanalysis data were obtained from the National Weather Service's National Centers for Environmental Prediction at

a gridded resolution of 0.25°×0.25° (ftp://arlftp.arlhq.noaa.gov/pub/archives/gfs0p25). The Euclidean distance clustering algorithm was chosen to cluster the air mass trajectories in the CB, DHS, and QL sites from June 20 to July 20, 2019. To obtain

the optimal clustering solution, the clustering result with the smallest percent change was selected (Wang et al., 2010).

PSCF is a gridded statistical analysis method that is used to combine the backward trajectory and the corresponding VOC concentration to determine the potential VOC source area (Liu et al., 2016b; Liu et al., 2020; Zheng et al., 2018). In this study, Xi'an City and the adjacent region covered by the back trajectories were divided into an array of 0.25°×0.25° grid cells. In this study, the pollution trajectory was defined as the trajectories corresponding to the total VOC (TVOC) concentration that

exceeded the 75[th] percentile concentration of TVOCs. When the number of endpoints of the pollution trajectory passing through the grid (i, j) is $M_{ij}$, and the number of endpoints of all the trajectories falling within the grid (i, j) is $N_{ij}$, then $PSCF_{ij}$ can be defined as the ratio of $M_{ij}$ to $N_{ij}$ (Polissar et al., 1999). The weight function $W_{ij}$ was used to increase the accuracy of the model, and $PSCF_{ij}$ can be calculated using Eq. (6) as follows:

$$PSCF_{ij} = \frac{M_{ij}}{N_{ij}} \times W_{ij} = \frac{M_{ij}}{N_{ij}} \times \begin{cases} 1.00 & N_{ij} > 80 \\ 0.70 & 40 < N_{ij} \le 80 \\ 0.42 & 5 < N_{ij} \le 40 \\ 0.05 & N_{ij} \le 5 \end{cases} \tag{6}$$

**2.6 Empirical Kinetic Modelling Approach**

The traditional empirical kinetic modelling approach (EKMA) is a model sensitivity tests of observation based box model and often used to evaluate the photochemical nonlinear relationship between ozone and precursors NOx and VOCs. The box model is based on the Regional Atmospheric Chemical Mechanisms version 2 (Goliff et al., 2013) updated with Leuven Isoprene Mechanism (Peeters et al., 2009). The definition and mechanism of the observation based box model is described elsewhere

in detail (Tan et al., 2017; Tan et al., 2018).

The EKMA curve can be used as a theoretical basis for designing emission reduction strategies to obtain the best ozone pollution reduction method (Jiang et al. 2018, Tan et al. 2018). The model input parameters include temperature, pressure, humidity, photolysis rate constant, $NO_2$, VOCs, etc. In this model, the ozone production rate P ($O_3$) is calculated by the ozone formation rate F ($O_3$) minus the ozone loss rate D ($O_3$), as shown in Eq. (7). The ozone formation rate F ($O_3$) and the ozone

loss rate D ($O_3$) can be calculated using Eq. (8) and (9) as follows:

$$P_{O_3} = F_{O_3} - D_{O_3} \tag{7}$$

$$F_{O_3} = k_{HO_2+NO}[HO_2][NO] + k_{(RO_2+NO)_{eff}}[RO_2][NO] \tag{8}$$

$$D_{O_3} = [O^1D][H_2O] + k_{O_3+OH}[O_3][OH] + k_{O_3+HO_2}[O_3][HO_2] + k_{O_3+alkenes}[O_3][alkenes]$$
$$+k_{OH+NO_2}[OH][NO_2] \tag{9}$$

where $k_{HO_2+NO}$ is represent rate coefficients for the reaction of the NO with $HO_2$ radical ($8.5 \times 10^{-12}$ molecule$^{-1}\cdot$cm$^3\cdot$s$^{-1}$, 298K); $k_{(RO_2+NO)_{eff}}$ is represent effective rate coefficients for the reaction of the NO with $RO_2$ radical ($8.5 \times 10^{-12}$ molecule$^{-1}\cdot$cm$^3\cdot$s$^{-1}$, 298K); $k_{O_3+OH}$ is represent rate coefficients for the reaction of the $O_3$ with OH radical ($7.3 \times 10^{-14}$ molecule$^{-1}\cdot$cm$^3\cdot$s$^{-1}$, 298K);

$k_{O_3+HO_2}$ is represent rate coefficients for the reaction of the $O_3$ with $HO_2$ radical ($1.9 \times 10^{-15}$ molecule$^{-1}$·cm$^3$·s$^{-1}$, 298K);

$k_{O_3+alkenes}$ is represent rate coefficients for the reaction of the $O_3$ with alkenes ($2.0 \times 10^{-17}$ molecule$^{-1}$·cm$^3$·s$^{-1}$, 298K);

$k_{OH+NO_2}$ is represent rate coefficients for the reaction of the $NO_2$ with OH radical ($1.1 \times 10^{-11}$ molecule$^{-1}$·cm$^3$·s$^{-1}$, 298K).

In this study, the average parameters of the entire observation period were used as the input parameters of the model to calculate the ozone concentration in the baseline scenario. Afterwards, the activity change array of VOCs and $NO_X$ were generated by changing each parameter in equal distance steps. The P ($O_3$) contours under these different VOCs and $NO_X$ reactivity conditions are called EKMA curves.

## 3 Results and discussion

### 3.1 Temporal and Spatial Characteristics of VOCs

### 3.1.1 Temporal variations

Temporal variations in wind speed, wind direction, temperature, relative humidity, $O_3$, and VOCs at the CB, DHS, and QL sites are shown in Figure 2. During the field observation campaign, 99 VOCs were measured, including 29 alkanes, 11 alkenes, 195 1 alkyne, 16 aromatics, 28 halohydrocarbons, 13 oxygenated VOCs (OVOCs), and 1 acetonitrile (Table S1). The CB and DHS sites were located in an urban area of Xi'an, while the QL site was located in a rural area. During the observation period, the average VOC concentrations at the CB, DHS, and QL sites were 27.8 ± 8.9, 33.8 ± 10.5, and 15.5±5.8 ppb, respectively. Due to the existence of more emission sources and lower wind speeds (0–2 m/s) in urban areas, the VOC concentrations were higher than those in rural areas. Overall, the VOC concentrations in Xi'an urban sites were approximately twice that of the 200 rural sites. The observation period occurred during summer in Xi'an, and the temperature was high, with average temperatures reaching 26.2±4.3 °C and 25.6±3.9 °C in urban and rural areas, respectively. Higher temperatures may accelerate a secondary transformation of VOCs into $O_3$, resulting in more frequent $O_3$ pollution incidents in Xi'an; the mean $O_3$ concentrations at the CB, DHS, and QL sites reached 50.2±29.9, 47.6±29.4, and 19.7±8.0 ppb, respectively.

During the observation period, there were two $O_3$ pollution events in the Xi'an urban area (CB and DHS sites), from June 23 205 to 26, 2019 and from June 30 to July 15, 2019. The wind speed diurnal pattern in the urban area displayed a single wave profile on both clean and polluted days with low wind speeds (0.4±0.2 m/s and 0.3±0.1 m/s, respectively) during the night and peaks (0.7±0.2 m/s and 0.9±0.3 m/s, respectively) at 14:00 CST. The overall wind speed in the Xi'an urban area was in a static state, indicating that the ability of wind to diffuse and dilute pollutants is limited (Figure 3). The variation trends of $O_3$ and temperature display a positive correlation, and the linear correlations between $O_3$ and temperature on polluted days ($R_{Pearson}$=0.7) 210 is stronger than that on clean days ($R_{Pearson}$=0.5). The value of temperature, $O_3$, and TVOCs all increased significantly on polluted days, indicating that the secondary transformation of VOCs to $O_3$ is more conducive at high temperatures. The difference between the VOC species concentration on clean and polluted days at the CB and DHS sites are shown in Figures 3c and 3e. As shown in Figure 3, isoprene concentrations at urban sites increased significantly during the $O_3$ pollution day,

which could due to the stronger plant emission at elevated temperature (Guenther et al., 1993; Guenther et al., 2012; Stavrakou et al., 2014). Concentrations of isoprene oxidation products (i.e., MVK and MACR) as well as most OVOCs also increased in the same period. However, similar concentrations of anthropogenic VOCs are found in clean and polluted days. This indicates a stronger photochemical conversion of VOCs existed in $O_3$ pollution days, which could due to the more favorable meteorological conditions (i.e., higher temperature and solar radiation). The specialty of OVOCs is that in addition to the primary emissions, OVOCs can also be formed through photochemical oxidation with alkenes and aromatics (Birdsall and Elrod 2011). The sources of OVOCs can be divided into anthropogenic primary sources, anthropogenic secondary sources, biogenic sources and background sources (Li et al., 2014; Wang et al., 2015). Base on the multi-linear regression model results (Figure S1 and S2) we found that the contribution of anthropogenic primary sources to OVOCs on $O_3$ pollution days is more significant.

### 3.1.2 Spatial variations

In this study, the VOC grid sampling was used to investigate the spatial variations in VOCs in Xi'an. A total of 20 sites were selected for grid monitoring that covered the entire city of Xi'an. Therefore, the results of the VOC grid sampling were used to represent the levels of the entire city. In the VOC grid sampling, 106 VOCs were measured, including 29 alkanes, 11 alkenes, 1 alkyne, 17 aromatics, 35 halohydrocarbons, 12 OVOCs, and carbon disulfide (Table S2-S3). The weather parameter on July 1 and July 14 in Xi'an is shown in Table S4. The average VOC concentration at the 20 sites on July 1 and July 14, 2019 was 29.1±8.4 ppb, and high VOC concentrations were clustered at the XF, CT, HC, ZYT, and YT sites (Figures 4a and 4b). Of the sites, XF site exhibited the highest VOC concentration of 54.0 ppb, followed by CT, HC, with concentrations of 41.4, and 38.2 ppb, respectively. XHT site exhibited the lowest concentration of VOCs at 18.0 ppb, followed by WQ and YST sites with concentrations of 19.6 ppb and 19.9 ppb, respectively. Compared with the results of the field observation campaign, the VOC concentration at the CB site was closer to the average concentration in Xi'an, and the VOC concentration at the DHS site was significantly higher than the average concentration. In terms of the VOC composition at each site, alkanes and OVOCs were dominant, accounting for 25.7–39.7% and 22.8–47.4% of the VOC concentration, respectively. In addition, the contribution of OVOCs at the YT site was significantly higher than that of the other sites, indicating that the YT site may be significantly affected by ageing sources (Figure 4a). The top 10 VOC species at the CB, DHS, QL, and grid sampling sites accounted for 66.1%, 63.4%, 71.1%, and 67.1% of the TVOC concentration, respectively (Table 2). Of these species, ethane, acetone, and propane were the top three contributors in Xi'an, accounting for 34.0–41.3% of the TVOC.

### 3.2 Sources

#### 3.2.1 Specific VOC Ratios

Different VOC species may have different sources; hence, the ratio of different species can be used to preliminarily analyze the difference in VOC sources at each site. The ratios of specific species that are often used are the toluene/benzene (T/B), m/p-xylene/ethylbenzene (X/E), iso-pentane/n-pentane, benzene/ethyne, toluene/ethyne, and m/p-xylene/ethyne ratios.

The T/B ratio is clearly different for various source profiles (Table S5). In industrial source emission researches, the ratio of T/B ranged from $1.4 \pm 0.8$ to $5.8 \pm 3.4$ by different industry type and process unit (Mo et al., 2015; Shi et al., 2015). In traffic source emission researches, the ratio of T/B ranged from $0.9 \pm 0.6$ to $2.2 \pm 0.5$ by different vehicle type and fuel composition (Qiao et al., 2012; Dai et al., 2013; Wang et al., 2013; Yao et al., 2013; Zhang et al., 2013; Yao et al., 2015b; Mo et al., 2016; Yao et al., 2015a; Deng et al., 2018). However, vehicle emissions include both diesel vehicles emissions and gasoline vehicles emissions in the atmospheric environment. Thus, the ratio of T/B in the traffic source should be closer to the results of the tunnel experiments which approximately $1.5 \pm 0.1$ (Liu et al., 2008a; Deng et al., 2018). In paint solvent usage source emission researches, the ratio of T/B was greater than $8.8 \pm 6.5$ by different solvent use process (Yuan et al., 2010; Wang et al., 2014; Zheng et al., 2013). In burning source emission researches, the T/B ratio was approximately $0.3 \pm 0.1$ in different combustion process and raw materials (Liu et al., 2008a; Li et al., 2011; Mo et al., 2016). In order to reduce the influence of photochemical reaction on the ratio of benzene to toluene, this study selected the weaker photochemical reaction period (3:00-7:00) for the analysis of toluene and benzene (Figure 5). Figure 5 shows that the ratios of toluene to benzene at the CB, DHS, QL, and gridded sampling sites were 1.1 ($R_{Pearson}=0.5$), 3.6 ($R_{Pearson}=0.6$), 0.5 ($R_{Pearson}=0.8$), and 1.75 ($R_{Pearson}=0.9$), respectively. In the urban areas (CB and DHS sites), most of the T/B ratios were distributed within the reference range of vehicle emissions and industrial emissions (Figure 5a, 5b), implying that vehicle sources and industrial sources contribute significantly to the VOCs in Xi'an urban area. In addition, the T/B value of some samples is greater than 5.8 in urban area which may affected by paint solvent usage source (Figure 5b). However, the detailed source contribution needs to be obtained through PMF source analysis results (Section 3.2.2). In the rural area (QL site), most of the T/B ratios were distributed within the reference range of vehicle emissions and burning emissions (Figure 5c), implying that vehicle sources and burning sources contribute significantly to the VOCs in Xi'an rural area. In the gridded sampling sites, the T/B ratio was predominately concentrated around 1.5, indicating that vehicle exhaust sources may greatly contributed to the overall VOCs in Xi'an (Figure 5d).

The reactivity of m/p-xylene with OH radicals was 2.7 times that of ethylbenzene (Carter, 2010); therefore, a lower X/E ratio represents a higher degree of air mass ageing. In areas with high air mass ageing, the contribution of external source transport to VOCs increased significantly. The diurnal variation of the X/E ratios at the CB, DHS, and QL sites demonstrates that the X/E ratio at the three sites all significantly decreased from 9:00 to 13:00 (Figures 6a–c). This indicated that there was a significant photochemical consumption effect on VOCs between 9:00 and 13:00. In addition, the X/E ratios at the CB and DHS sites significantly increased after 13:00, while variations in the X/E ratio were not clear at the QL site, indicating that there were more primary emissions from anthropogenic sources at the urban sites (CB and DHS sites). The diurnal variations

in OH exposure exhibit an inverse correlation with the X/E ratio, reaching a maximum daily value between 12:00–15:00
(Figures 6d–f). The average OH exposures of the CB, DHS, and QL sites were $5.1\times10^{10}$, $1.7\times10^{10}$, and $3.1\times10^{10}$ molecule cm$^{-3}$ s, respectively.

The ratio of iso-pentane/n-pentane is clearly different for various sources. Recent studies have found that the iso-pentane/n-pentane ratio was 2.93 for vehicle exhaust sources (Liu et al., 2008a) and 0.56–0.8 for coal burning (Yan et al., 2017). In this study, linear correlation coefficients between iso-pentane and n-pentane at the CB, DHS, QL, and gridded sampling sites were
280 1.8 ($R_{Pearson}=0.7$), 1.1 ($R_{Pearson}=0.8$), 1.5 ($R_{Pearson}=0.9$), and 3.2 ($R_{Pearson}=1.0$), respectively (Figure 7a). These results indicated that propane sources in Xi'an are greatly affected by vehicle emissions. Propane and ethane are the main components of liquefied petroleum gas (LPG) and natural gas (NG) (Blake and Rowland, 1995; Katzenstein et al., 2003). The propane/ethane (P/E) ratio in LPG vehicle exhaust was approximately 3, which is significantly higher than that in gasoline and diesel vehicles (Ho et al., 2009). Therefore, the P/E ratio is often used to indicate the impacts of LPG and NG use on VOC concentration. In
this study, the linear correlation coefficients between propane and ethane at the CB, DHS, QL, and gridded sampling sites were 0.8 ($R_{Pearson}=0.8$), 0.6 ($R_{Pearson}=0.7$), 1.0 ($R_{Pearson}=0.9$), and 0.5 ($R_{Pearson}=0.5$), respectively (Figure 7b). Xi'an buses and taxis use dual fuels (gasoline and natural gas); therefore, the P/E ratio in Xi'an is significantly lower than that in Guangzhou (1.27), where some buses and taxis use LPG fuel (Tsai et al., 2006; Wang et al., 2018; Yuan et al., 2012).

### 3.2.2 Sources apportionment

During the observation period (June 20, 2019 to July 20, 2019), seven-factor PMF solutions were selected for the CB, DHS, and QL sites, and nine possible emission sources were identified: vehicle exhaust, industrial sources, combustion, paint solvent usage, biogenic sources, fuel evaporation, biogenic burning, ageing sources, and vehicle exhaust + industrial sources. The source profiles and contributions to VOC concentrations in the CB, DHS, and QL sites during the observation period are displayed in Figure 8. The relationships between wind direction, wind speed, and the main sources of the three stations are
illustrated in Figure 9.

Vehicle exhaust sources are characterized by high concentrations of alkanes (Cai et al., 2010), a few alkenes (An et al., 2017; Liu et al., 2008a), ethyne (Ling et al., 2011), and $C_6$–$C_7$ aromatics (Mo et al., 2018). In this study, vehicle exhaust sources in the CB and QL sites exhibited high ethane, propane, butane, and acetone contents. In the DHS site, vehicle exhaust sources exhibited high ethane, propane, butane, pentane, ethylene, ethyne, and acetone contents and a few aromatics. These species
are important vehicle exhaust tracers. Thus, these factors are identified as sources of vehicle exhaust. At the end of 2019, the number of motor vehicles in Xi'an city exceeded 3.43 million, with the city ranking 8th in China for motor vehicle number (https://www.mps.gov.cn). Moreover, the contribution of vehicle exhaust to VOCs at the CB, DHS, and QL sites were 41.3%, 30.6%, and 23.6–41.4% (Figure 8), respectively. Vehicle exhaust has become an important source of VOCs emissions in Xi'an. The CPF results in Figure 9 indicated that vehicle exhaust at the CB site had a high potential (CPF > 0.6) of source transport
from the southwest when the wind speed exceeded 1 m/s. However, the vehicle exhaust at the DHS and QL sites were not significantly affected by the transmission source (CPF < 0.3).

Industrial sources are characterized by high concentrations of halohydrocarbons (Dumanoglu et al., 2014; Sun et al., 2016) and aromatics (Guo et al., 2011). They also contain alkanes and alkenes, such as ethane, hexane, and ethylene (Liu et al., 2008a), which are important raw materials for industrial production. Factors that meet these characteristics were identified in this study as industrial sources. The industrial sources of the CB site contained a higher proportion of n-undecane and n-dodecane, which are important tracers for asphalt industry applications (Liu et al., 2008a). The source contribution results demonstrate that industrial emissions are the second-largest source of VOC emissions in the urban areas of Xi'an, accounting for 29.7% and 30.6% at the CB and DHS sites, respectively (Figure 8). Because the QL site is located in a rural area with few factories, the contribution of industrial sources is small. The CPF results (Figure 9) indicated that industrial source at the CB and DHS sites were not significantly affected by the transmission source (CPF < 0.3).

Combustion sources are characterized by high concentrations of ethane, ethylene, propane, and ethyne (Ling et al., 2011; Liu et al., 2008a; Song et al., 2018). Factors that meet these characteristics were identified as combustion sources in this study. Combustion sources have become the third-largest source of VOCs in Xi'an, accounting for 10.9%, 12.3%, and 12.8% at the CB, DHS, and QL sites, respectively (Figure 8). The CPF results demonstrated that combustion sources at the QL site exhibited a high potential (CPF > 0.8) of source transport from the north when the wind speed exceeded 3 m/s (Figure 9).

Paint solvent usage sources are characterized by high concentrations of aromatics, such as toluene, ethylbenzene, m/p-xylene, and o-xylene (An et al., 2017; Li et al., 2018; Song et al., 2019). Factors that meet these characteristics were identified as paint solvent usage sources in this study. Paint solvent usage sources contributed significantly to the VOCs in Xi'an, accounting for 8.2%, 12.1%, and 11.2% at the CB, DHS, and QL sites, respectively (Figure 8), which was comparable to the contribution of combustion sources.

Biogenic sources are characterized by high concentrations of isoprene and the oxidation products of isoprene (methacrolein and methyl vinyl ketone) (Gong et al., 2018; Ling and Guo, 2014; Ling et al., 2019). Factors that meet these characteristics were identified as biogenic sources in this study. Biogenic sources were identified at the DHS and CB sites, accounting for 7.2% in both sites (Figure 8).

Biogenic burning sources are characterized by high concentrations of ethyne, methyl chloride, benzene, and toluene (Liu et al., 2008a). The fifth factor of the QL site met this characteristic and was identified as a biogenic burning source, accounting for 8.2% (Figure 8).

Fuel evaporation sources are characterized by high concentrations of iso-butane, n-butane, iso-pentane, n-pentane, 2-methylpentane, and 3-methylpentane (Liu et al., 2017; Zheng et al., 2020). The seventh factor of the CB site met this characteristic and is identified as a fuel evaporation source, accounting for 9.8% (Figure 8).

Ageing sources are characterized by high concentrations of OVOCs (Li et al., 2015; Zhu et al., 2018). As important tracers of ageing sources, OVOCs include both primary and secondary sources and have a longer lifetime in the atmosphere (Derstroff et al., 2017). The fourth factor of the QL site met this characteristic and was identified as an ageing source, accounting for 19.2% (Figure 8). The CPF results illustrate that ageing sources at the QL site exhibited a high potential (CPF > 0.8) of source transport from the east when the wind speed exceeded 1 m/s (Figure 9).

## 3.3 Cluster and PSCF results

The 24-h backward trajectories from Xi'an for the cluster and PSCF analysis are shown in Figure 10. Based on the figure, air mass back trajectories can be clustered into the eastern trajectories (Cluster 1), southeastern trajectories (Cluster 2+4), and northeastern trajectories (Cluster 3+5) at the CB site (Figure 10a). It is evident that the proportion of southeastern trajectories to the total trajectories and that of the southeastern pollution trajectories to the total pollution trajectories were significantly higher than those of the other cluster trajectories, accounting for 58.7% and 60.8%, respectively (Table 3). There were two trajectory clusters from the southeast direction, the southeast short distance trajectories (Cluster 2) and southeast medium-long distance trajectories (Cluster 4), accounting for 35.2% and 23.5%, respectively. This result indicated that the VOC concentration in the CB site was significantly affected by the southeast trajectory from the junction of the Shaanxi Province, Hubei Province, and Henan Province in addition to local sources. In addition, although the proportion pollution trajectories from the northwest (LDT) was small (Cluster 3), the concentrations in these pollution trajectories were the greatest, reaching 41.5 ppb (Table 3). Thus, attention should be paid to the long-distance transmission of highly polluting air masses from Inner Mongolia. Regarding the VOC composition (Figure 10b), alkanes with lower activities accounted for a larger proportion in the long-distance trajectories (Cluster 3).

The air mass back trajectories can be clustered into the eastern trajectories (Cluster 1), southeastern trajectories (Cluster 2+4), and northeastern trajectories (Cluster 3+5) at the DHS site (Figure 10c). It is evident that the proportion of southeast trajectories to the total trajectories and that of the southeast pollution trajectories to the total pollution trajectories were significantly higher than those of the other cluster trajectories, accounting for 68.9% and 73.3%, respectively (Table 3). This result indicated that the VOC concentration in the DHS site was mainly affected by pollution transmission at the Shaanxi Province, Hubei Province, and Henan Province junction and local source emissions. In addition, although the proportion of pollution trajectories from the northwest (LDT) was small (Cluster 3), the concentrations in these pollution trajectories were the greatest, reaching 57.9 ppb. Thus, attention should be paid to the long-distance transmission of highly polluting air masses from central Inner Mongolia.

The air mass back trajectories can be clustered into the eastern trajectories (Cluster 1), northwestern trajectories (Cluster 2), southeastern trajectories (Cluster 3+4), and northeastern trajectories (Cluster 5) at the QL site (Figure 10e). It is evident that the proportion of northeastern trajectories to the total trajectories and that of the northeastern pollution trajectories to the total pollution trajectories were significantly higher than those of the other cluster trajectories, accounting for 36.4% and 34.3%, respectively (Table 3). However, the VOC concentrations of the eastern trajectories to the total trajectories were significantly higher than those of the other cluster trajectories, reaching 18.7 ppb. This result indicated that the VOC concentration in the QL site was significantly affected by the east trajectory from the junction of the Shanxi Province, Shaanxi Province, and Henan Province in addition to local sources.

Based on the PSCF analysis for Xi'an (Figures 10a, 10c, and 10e), in the urban sites, high PSCF values were mainly observed to the east and south of Xi'an, and in rural sites, the high PSCF values were primarily observed to the east of Xi'an. Different air mass tracking time (6h, 12h, 24h) were used in the PSCF analysis of different VOC species (Figure S3). Strong chemically

active species (e.g., ethylene and xylene) had shorter air mass tracks, with high PSCF values appeared in areas near sites.

However, the high PSCF values of long lifetime species (acetone) were found not only near the site but also in the eastern and southern regions of the site. The highest PSCF values of TVOCs appeared in areas near the CB, DHS, and QL sites, which indicated that Xi'an has a strong local source.

### 3.4 Secondary Transformation Potential

### 3.4.1 $L_{OH}$ and OFP

The VOC loss rates ($L_{OH}$) that react with OH at the CB, DHS, and QL sites were 4.9 $s^{-1}$, 4.8 $s^{-1}$, and 2.4 $s^{-1}$, respectively. The average $L_{OH}$ of the VOCs at the 20 gridded sites was 5.0 $s^{-1}$, and high VOC $L_{OH}$ were concentrated at the XF, ZYT, CT, LTC, and LT sites (Figure 11a). Of the sites, XF exhibited the highest $L_{OH}$ of 16.2 $s^{-1}$, followed by ZYT and CT with $L_{OH}$ of 9.9 $s^{-1}$ and 6.6 $s^{-1}$, respectively. The XX site exhibited the lowest $L_{OH}$ of 1.6 $s^{-1}$, followed by the WQ and YST sites with $L_{OH}$ of 1.6 $s^{-1}$ and 2.3 $s^{-1}$, respectively. At the CB, DHS, and QL sites, alkenes and OVOCs played a dominant role in $L_{OH}$, accounting for

31.7–50.8% and 26–35.8% (Figure 11a). For the gridded sampling sites, the overall $L_{OH}$ in Xi'an was significantly affected by alkenes, accounting for 45.4–85.9%. The top 10 VOC species of $L_{OH}$ at the CB, DHS, QL, and grid sampling sites accounted for 69.2%, 60.8%, 75.9%, and 80.2% of the total $L_{OH}$, respectively (Table 2). Of these species, isoprene, acetaldehyde, ethylene, and 1-butene were the greatest contributors to total $L_{OH}$ in Xi'an, reaching 36.4–63.0%.

The $O_3$ formation potentials of VOCs at the CB, DHS, and QL sites were 66.9, 74, and 35.6 ppb, respectively. The average

VOC OFP at the 20 gridded sites was 53.9 ppb, and the high VOC OFPs were concentrated at the XF, ZYT, LTC, CT, and HC sites (Figure 11b). Of the sites, XF exhibited the highest OFP of 140.7 ppb, followed by ZYT and LTC with VOC OFPs of 87.9 and 75.7 ppb, respectively. The WQ site exhibited the lowest OFP of 21.9 ppb, followed by the XX and XHT sites with OFPs of 30.76 and 30.79 ppb, respectively. At the CB, DHS, and QL sites, alkenes and OVOCs played a dominant role in the OFP, accounting for 30.1–40.2% and 31.1–39.4%, respectively (Figure 11b). For the gridded sampling sites, the overall

OFP in Xi'an was significantly affected by alkenes and aromatics, accounting for 39.7–67.7% and 7.2–23.2%, respectively. The top 10 VOC OFP species at the CB, DHS, QL, and grid sampling sites accounted for 67.3%, 66.8%, 73.3%, and 73.3% of the total OFP, respectively (Table 2). Of these species, ethylene, acetaldehyde, isoprene, and m/p-xylene were the greatest contributors in Xi'an, reaching 44.2–54.5%.

### 3.4.2 VOCs-NOx-$O_3$ Sensitivity

The relationship between the ozone production rates (P ($O_3$)), anthropogenic VOCs (AVOCs) reactivity and NOx reactivity of the CB, DHS, and QL sites during the observation period was shown in Figure 12. The black curve in the Figure 12 represents the P ($O_3$) contour, and the black straight line represents the connection line of the P ($O_3$) turning point (ridgeline), whose slope represents the photochemical parameter $k_{NOx}/k_{AVOCs}$ (Jiang et al., 2018). When the site's $k_{NOx}/k_{AVOCs}$ value is located above the ridgeline, it means that ozone formation is under VOCs-limited regime, otherwise it means that ozone formation is under NOx-

limited regime. It can be seen from Figure 12 that the ozone generation of QL site is located in the NOx-limited regime, and reducing NOx can effectively control ozone generation. The ozone generation of DHS site is located in the VOCs-limited regime, and reducing VOCs can effectively control ozone generation. However, CB site is located in the transition regime between VOC- and NOx-limited regimes. Therefore, simultaneous reduction of VOCs and NOx concentration should be considered at CB site to achieve the purpose of controlling $O_3$.

## 4 Conclusions

In this study, a campaign of field observations and VOC grid sampling was conducted in Xi'an from June 20 to July 20, 2019. During the observation period, the average VOC concentrations at the CB, DHS, QL and gridded sampling sites were 27.8 ± 8.9, 33.8 ± 10.5, 15.5±5.8, and 29.1±8.4 ppb, respectively. Overall, the concentrations of VOCs in the Xi'an urban sites were approximately twice those of the rural sites. Due to a lower diffusion capacity and higher conversion capacity, the $O_3$ concentration in Xi'an urban areas (CB and DHS) often exceeded the national hourly standard of 200 μg/m$^3$ (approximately 101.9 ppb). In terms of the composition of VOCs at each site, alkanes and OVOCs were dominant in the VOC concentration, accounting for 25.7–39.7% and 22.8–47.4% of the TVOCs, respectively.

The PMF results demonstrated that vehicle exhaust, industry, combustion, paint solvent usage, and ageing are the primary sources of VOCs in Xi'an. Of the sources, vehicle exhaust was the primary source of VOC emissions in Xi'an, and the contributions of vehicle exhaust to the VOCs at the CB, DHS, and QL sites were 41.3%, 30.6%, and 23.6–41.4%, respectively. In urban areas, industrial emissions were the second-largest source of VOC emissions, accounting for 29.7% and 30.6% at the CB and DHS sites, respectively. However, the contribution of ageing sources was greater in rural areas, accounting for 19.2% at the QL site.

Based on the 24-h backward trajectories and PSCF analysis for Xi'an, in the urban areas, the VOC concentration was significantly affected by the southeast trajectory, and the high PSCF values were primarily observed to the east and south of Xi'an. In rural areas, the VOC concentration was significantly affected by the northeast trajectory, and the high PSCF values were primarily observed to the east of Xi'an. The highest PSCF values appeared in the area near the CB, DHS, and QL sites, which indicates that Xi'an has a strong local source.

In Xi'an, alkenes, aromatics, and OVOCs played a dominant role in secondary transformation. Isoprene, acetaldehyde, ethylene, and 1-butene were the greatest contributors to $L_{OH}$ in Xi'an, reaching 36.4–63.0%. Ethylene, acetaldehyde, isoprene, and m/p-xylene were the greatest contributors to OFP in Xi'an, reaching 44.2–54.5%. These are the major species of concern for reducing $O_3$ pollution in Xi'an. The VOCs-NOx-$O_3$ sensitivity analysis results showed that the ozone generation of DHS site is located in the VOCs-limited regime, CB in the transition regime between VOC- and NOx-limited regimes, and QL sites is located in the NOx-limited regime. Therefore, reducing VOCs concentration at DHS site, reducing VOCs and NOx concentration at CB site, and reducing NOx concentration at QL site can effectively control ozone generation.

## Data availability

The underlying research data can be accessed upon contact with the corresponding author (Xin Li: li_xin@pku.edu.cn).

## Author contribution

XL and YZ designed the study. MS and XL analyzed the data and wrote the paper. XL, LZ, HD, KL, SC, JC and NZ planned
the locations of field observations and VOC grid sampling sites and built field observations sites. MS, SY, XY, YY and SZ contributed to field observations and preprocessed data. KL, QC, XY contributed to VOC grid sampling. MS and XL contributed to revise the paper. All authors contributed to discussed, and improved the paper.

## Competing interests

The authors declare that they have no conflict of interest.

## Acknowledgements

This work was supported by the National Natural Science Foundation of China (91644108, 91844301) and by the National Key R&D Program of China (2017YFC0209400).

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

**Table 1: Detailed sampling site information.**

| Site name | Abbreviation | Latitude | Longitude | Parameter |
|---|---|---|---|---|
| Chanba | CB | N34°20'12" | E109°01'35" | O₃/NO/NO₂/NOx/CO/ |
| Di Huan Suo | DHS | N34°13'49" | E108°52'58" | photolysis rate/VOCs (99)/ |
| Qinling | QL | N34°04'11" | E108°20'31" | T/P/RH |
| Caotan | CT | N34°23'4" | E108°52'13" | |
| College of Automotive Technology | CAT | N34°12'48" | E109°5'54" | |
| Gengzhen Sub-district Office | GZ | N34°26'52" | E109°7'31" | |
| Guang Yun Lake | GYL | N34°19'23" | E109°3'1" | |
| Hu County | HC | N34°6'43" | E108°36" | |
| Jiaodai Town | JDT | N34°3'33" | E109°14'39" | |
| Jiufen Town | JFT | N34°3'35" | E108°25'50" | |
| Lantian County | LTC | N34°9'9" | E109°19'5" | |
| Luan Town | LT | N33°59'14" | E108°50'4" | |
| Region Site | RS | N34°10'45" | E108°44'18" | VOCs (106) |
| Wangqu Sub-district Office | WQ | N34°4'36" | E108°59'54" | |
| Xiaoyan | XY | N34°13'4" | E108°56'15" | |
| Xiehu Town | XHT | N34°13'47" | E109°13'41" | |
| Xinfeng Sub-district Office | XF | N34°25'23" | E109°16'12" | |
| Xinshi Sub-district Office | XS | N34°30'38" | E109°11'7" | |
| Xinxing Sub-district Office | XX | N34°39'43" | E109°16'29" | |
| Yin Town | YT | N34°0'17" | E109°6"44" | |
| Yushan Town | YST | N34°12'53" | E109°31'33" | |
| Zhouzhi County | ZZC | N34°9'35" | E108°12'33" | |
| Zhuyu Town | ZYT | N34°5'50" | E108°7'41" | |

Note. T = temperature, P = pressure, RH = humidity.

**Table 2: Top 10 VOC concentration, $L_{OH}$, and OFP species in the CB, DHS, QL, and grid sampling sites.**

| Site | Rank | Species | Concentration (ppb) | Species | $L_{OH}$ ($\times 10^{-1}$/s) | Species | OFP (ppb) |
|------|------|---------|---------------------|---------|----------------|---------|-----------|
| CB | 1 | Acetone | 4.9 | Isoprene | 13.8 | Ethylene | 11.8 |
| | 2 | Ethane | 3.0 | Acetaldehyde | 5.8 | Acetaldehyde | 10.2 |
| | 3 | Propane | 1.9 | Ethylene | 2.7 | Isoprene | 6.0 |
| | 4 | Acetaldehyde | 1.6 | m/p-Xylene | 2.2 | m/p-Xylene | 3.8 |
| | 5 | Ethyne | 1.4 | trans-2-Butene | 2.0 | Methylvinylketone | 2.6 |
| | 6 | Ethylene | 1.3 | 1-Butene | 1.8 | 1-Butene | 2.3 |
| | 7 | n-Butane | 1.3 | Propanal | 1.4 | Propene | 2.2 |
| | 8 | Dichloromethane | 1.2 | n-Hexanal | 1.4 | Propanal | 2.1 |
| | 9 | Isobutane | 1.0 | 1-Hexene | 1.4 | Toluene | 2.1 |
| | 10 | Isopentane | 0.9 | Methylvinylketone | 1.3 | trans-2-Butene | 1.9 |
| DHS | 1 | Acetone | 5.6 | Isoprene | 7.0 | Ethylene | 12.9 |
| | 2 | Ethane | 3.9 | Acetaldehyde | 6.6 | Acetaldehyde | 11.8 |
| | 3 | Propane | 2.1 | Ethylene | 3.0 | m/p-Xylene | 5.0 |
| | 4 | Acetaldehyde | 1.8 | m/p-Xylene | 3.0 | Methylvinylketone | 4.0 |
| | 5 | Dichloromethane | 1.7 | Methylvinylketone | 2.0 | Toluene | 3.4 |
| | 6 | n-Butane | 1.6 | Propanal | 1.6 | Isoprene | 3.0 |
| | 7 | Ethyne | 1.5 | Propene | 1.6 | Propene | 2.9 |
| | 8 | Ethylene | 1.4 | Styrene | 1.6 | Propanal | 2.4 |
| | 9 | Isopentane | 1.0 | Dodecane | 1.4 | Acetone | 2.0 |
| | 10 | Freon11 | 0.9 | Methacrolein | 1.3 | o-Xylene | 2.0 |
| QL | 1 | Ethane | 2.2 | Isoprene | 6.7 | Acetaldehyde | 8.7 |
| | 2 | Acetone | 2.1 | Acetaldehyde | 4.9 | Ethylene | 6.6 |
| | 3 | Propane | 1.4 | Ethylene | 1.5 | Isoprene | 2.9 |
| | 4 | Acetaldehyde | 1.3 | Propene | 1.0 | Propene | 1.8 |
| | 5 | Ethyne | 1.0 | n-Hexanal | 0.9 | Methylvinylketone | 1.3 |
| | 6 | n-Butane | 0.7 | m/p-Xylene | 0.7 | m/p-Xylene | 1.2 |
| | 7 | Ethylene | 0.7 | 1-Butene | 0.7 | Ethyne | 0.9 |
| | 8 | Freon11 | 0.7 | Methylvinylketone | 0.6 | 1-Butene | 0.9 |
| | 9 | Dichloromethane | 0.5 | Methacrolein | 0.6 | Propanal | 0.9 |
| | 10 | Isopentane | 0.3 | Propanal | 0.6 | n-Butane | 0.9 |
| Xi'an | 1 | Acetone | 6.6 | Isoprene | 27.1 | Isoprene | 11.7 |
| | 2 | Ethane | 3.4 | Ethylene | 2.4 | Ethylene | 10.1 |
| | 3 | Propane | 2.1 | 1-Butene | 2.1 | m/p-Xylene | 2.9 |

| 4 | n-Butane | 1.3 | m/p-Xylene | 1.7 | Propene | 2.9 |
|---|---|---|---|---|---|---|
| 5 | Ethyne | 1.3 | Propene | 1.6 | 1-Butene | 2.6 |
| 6 | Isopentane | 1.2 | MTBE | 1.4 | Acetone | 2.4 |
| 7 | Ethylene | 1.1 | Acrolein | 1.2 | Toluene | 1.9 |
| 8 | Isoprene | 1.1 | Isopentane | 1.0 | Acrolein | 1.8 |
| 9 | Dichloromethane | 0.8 | trans-2-Butene | 0.9 | Isopentane | 1.7 |
| 10 | Freon11 | 0.7 | n-Butane | 0.8 | n-Butane | 1.5 |

**Table 3: Air mass cluster analysis results for the CB, DHS, and QL sites.**

| Site | Cluster | Air mass origins | Ratio (%) | TVOCs (ppb) | P_Ratio (%) | P_TVOCs (ppb) |
|------|---------|------------------|-----------|-------------|-------------|----------------|
| | 1 | East | 10.3 | 27.9 | 8.4 | 38.3 |
| | 2 | Southeast | 35.2 | 29.3 | 42.8 | 39.6 |
| CB | 3 | Northwest (LDT) | 15.2 | 26.1 | 8.4 | 41.5 |
| | 4 | Southeast (M-LDT) | 23.5 | 25.7 | 18.1 | 40.5 |
| | 5 | Northeast | 15.7 | 29.5 | 22.3 | 40.3 |
| | 1 | East | 12.1 | 34.9 | 11.9 | 45.2 |
| | 2 | Southeast | 42.5 | 36.7 | 58.5 | 48.2 |
| DHS | 3 | Northwest (LDT) | 6.7 | 30.2 | 4.5 | 57.9 |
| | 4 | Southeast (M-LDT) | 26.4 | 31.4 | 14.8 | 45.2 |
| | 5 | Northwest | 12.4 | 32.5 | 10.2 | 48.3 |
| | 1 | East | 13.4 | 18.7 | 20.0 | 23.7 |
| | 2 | Northwest (LDT) | 16.7 | 13.3 | 9.1 | 22.7 |
| QL | 3 | Southeast | 18 | 15.2 | 17.1 | 22.9 |
| | 4 | Southeast (M-LDT) | 15.5 | 15.4 | 19.4 | 23.8 |
| | 5 | Northeast | 36.4 | 15.3 | 34.3 | 23.3 |

Note: LDT means long distance trajectory. M-LDT means medium-long distance trajectory. P_Ratio means the proportions of the pollution trajectories in each type of clustering trajectory. P_TVOCs means the average TVOC concentration of the pollution trajectories.


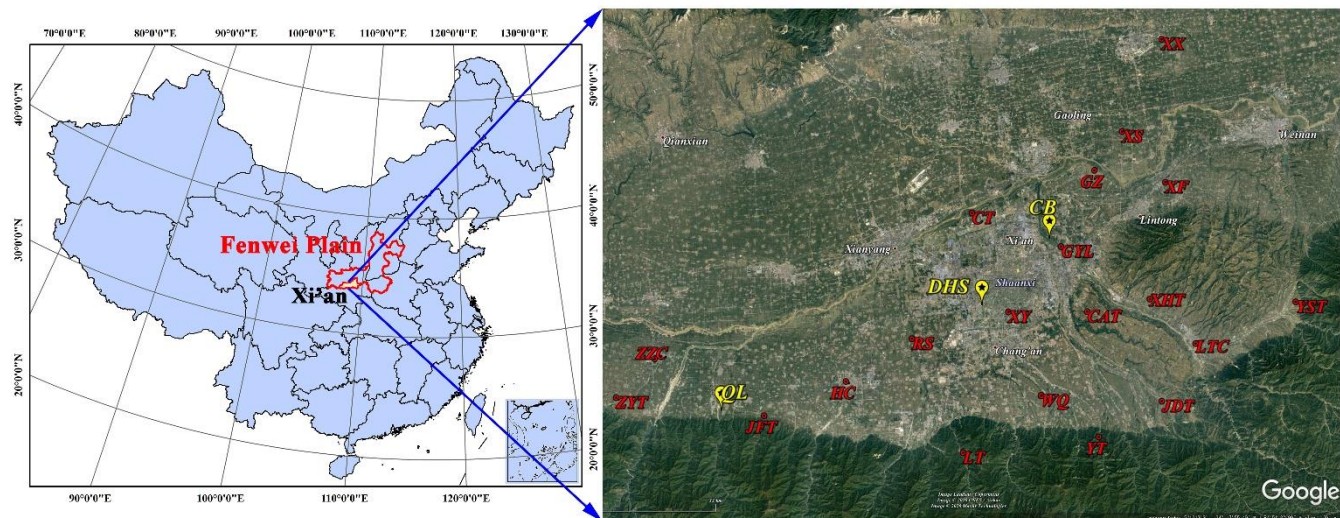

**Figure 1: VOC field observation and grid sampling sites in Xi'an (© Google Earth).**

Note: The topographic image was obtained from Google Earth.

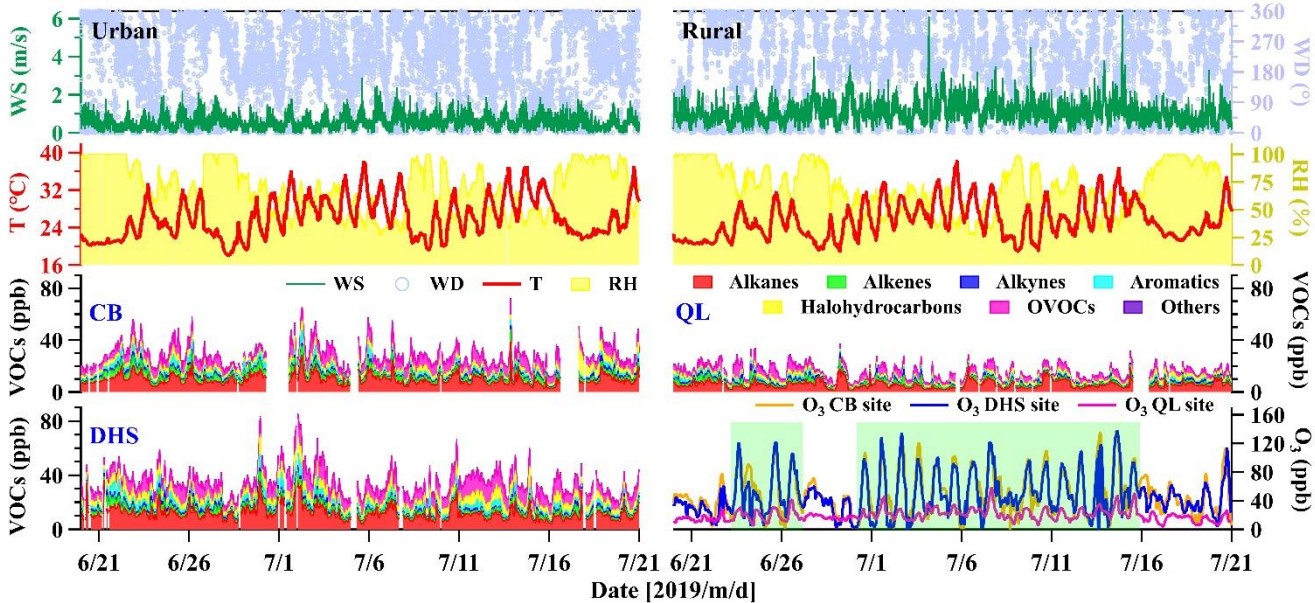

**Figure 2: Time series of wind speed (WS), wind direction (WD), temperature (T), relative humidity (RH), O₃, and VOCs at the CB, DHS, and QL sites.**

Note: The green area in the figure represents periods with $O_3$ levels exceeding the Chinese national standard at the CB and DHS sites. Time is expressed in China CST.

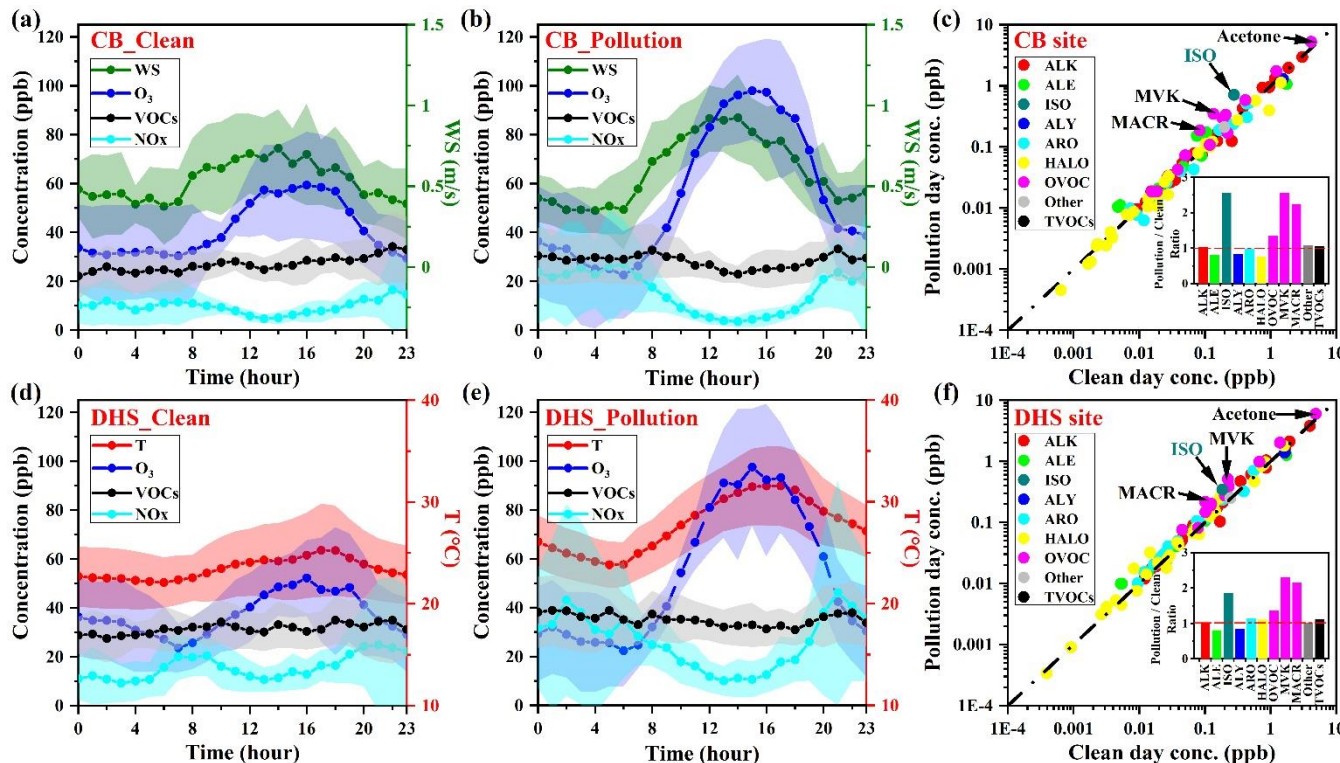

**Figure 3:** Diurnal variations in wind speed (WS), temperature (T), $O_3$, NOx, and TVOCs on clean and polluted days at the (a) and (b) CB and (d) and (e) DHS sites. Differences in VOC concentrations between clean and polluted days at the (c) CB and (f) DHS sites.

Note: ALK = alkanes, ALE = alkenes (except isoprene), ISO = isoprene, ALY = alkynes, ARO = aromatics, and HALO = Halohydrocarbons, MVK = Methyl Vinyl Ketone, and MACR = Methacrolein.

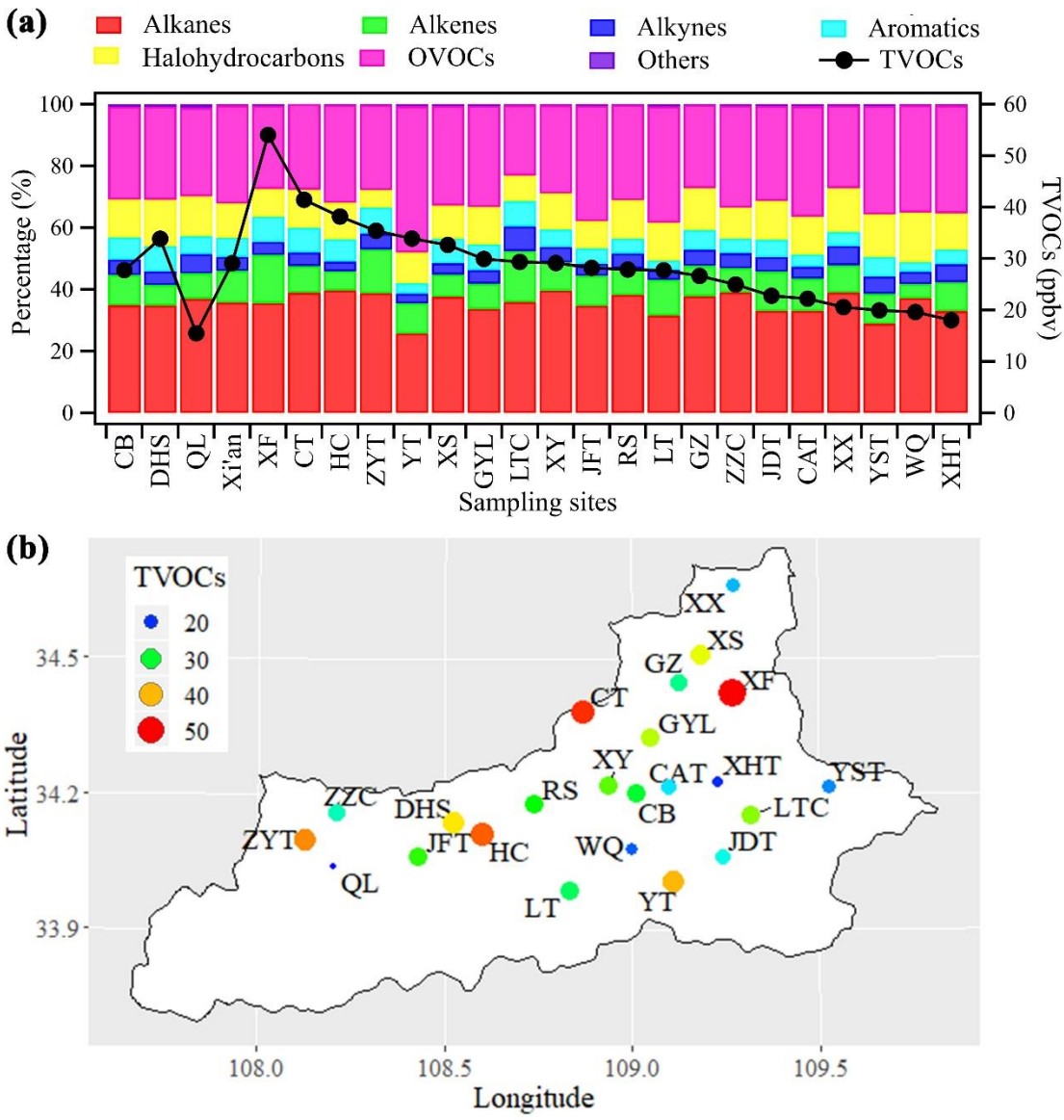

**Figure 4: (a) Proportions of seven VOCs groups and averaged TVOCs concentrations in different sites in Xi'an. (b) Spatial distribution of averaged TVOC concentrations in different sites in Xi'an.**

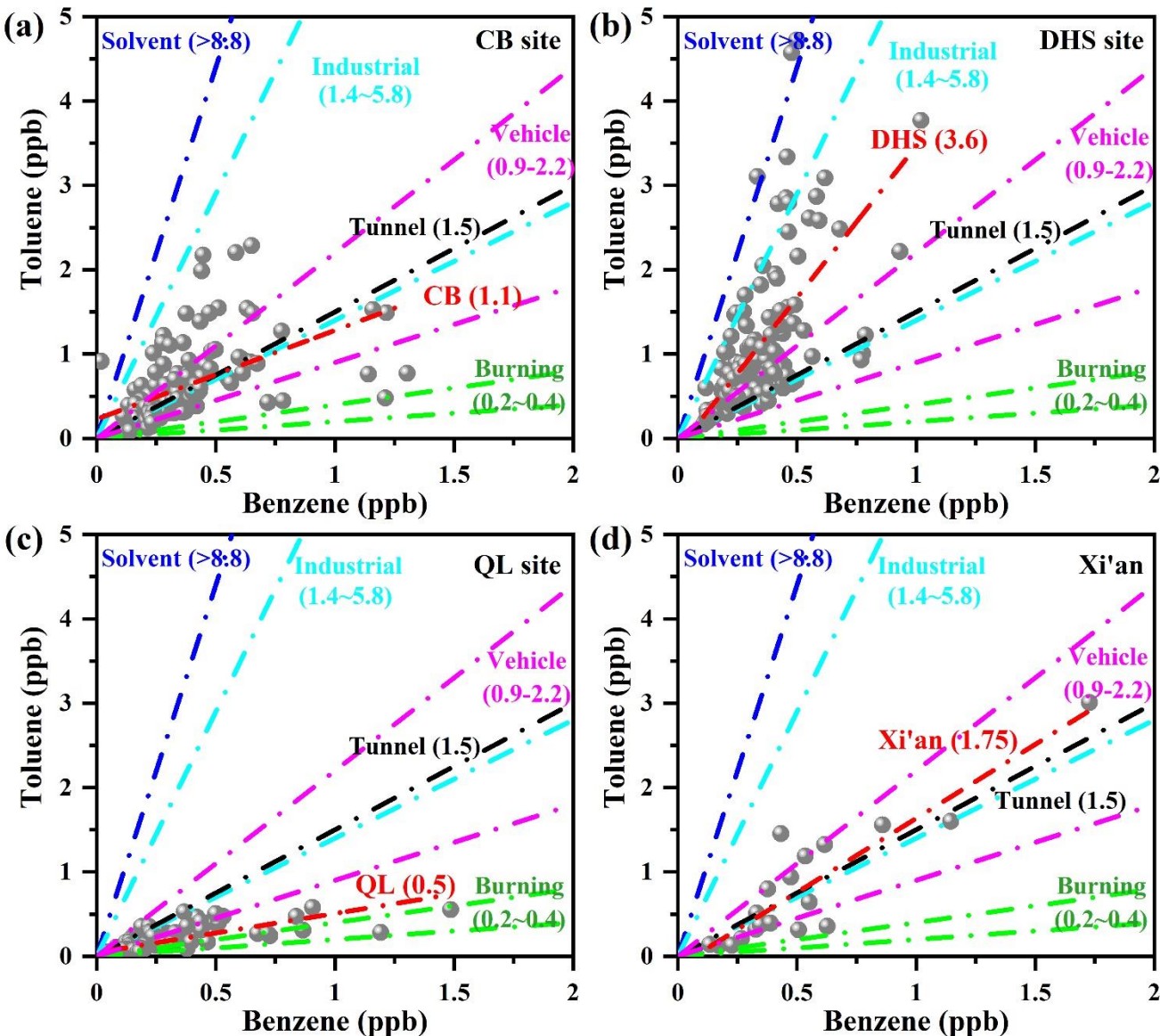

Figure 5: Linear correlations between toluene and benzene at the CB, DHS, QL, and gridded sampling sites between 3:00-7:00 during the observation period.


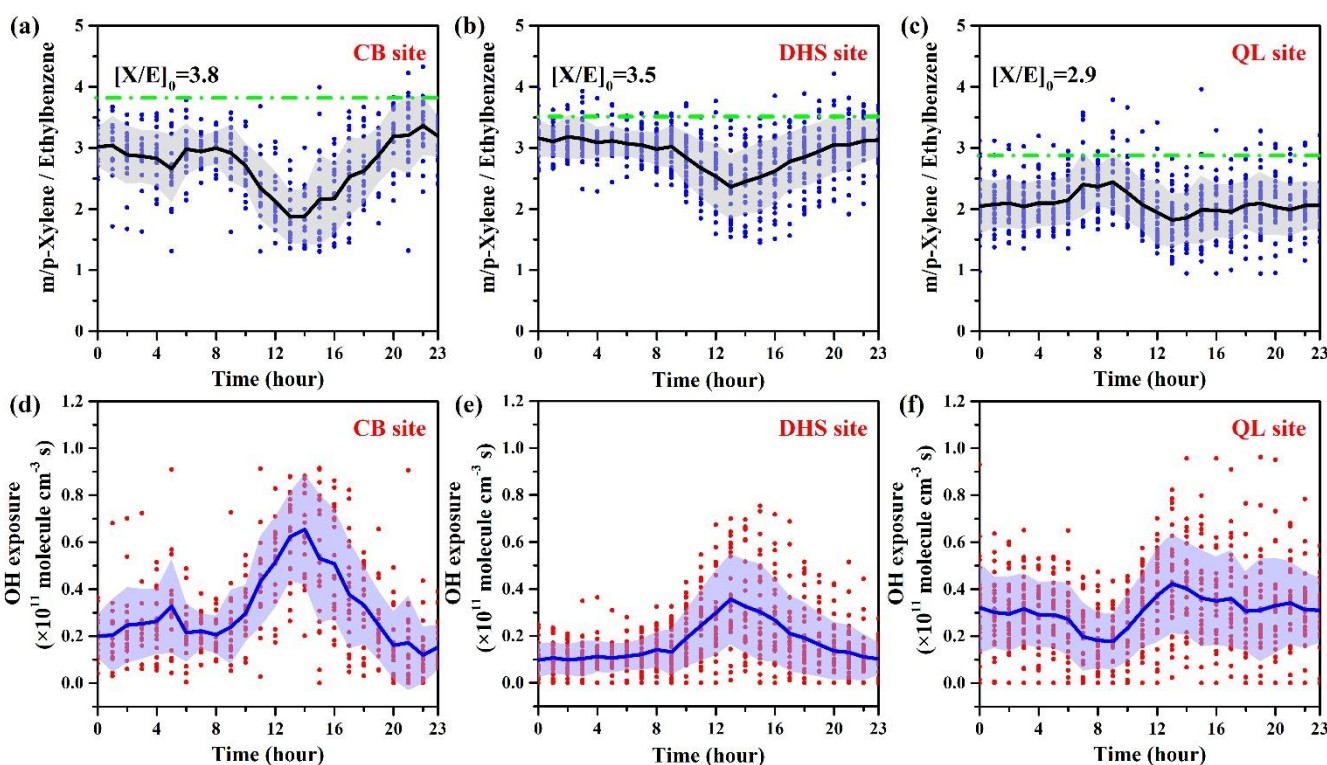

**Figure 6: Diurnal variations in m/p-xylene to ethylbenzene and OH exposure at the CB, DHS, QL, and gridded sampling sites.**

Note: Time is expressed in CST. The green line represents the initial emission ratio of m/p-xylene and ethylbenzene.

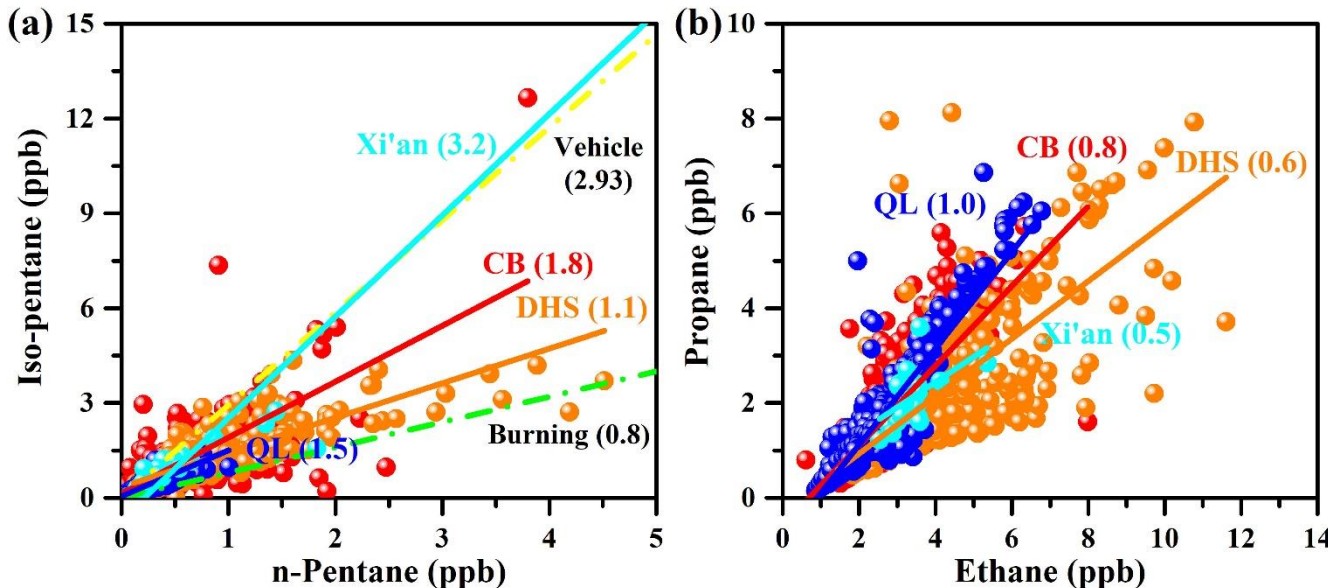


**Figure 7: Linear correlations ($R^2$) between (a) iso-pentane and n-pentane and (b) propane and ethane at the CB (red), DHS (orange), QL (blue), and gridded sampling sites (light blue).**

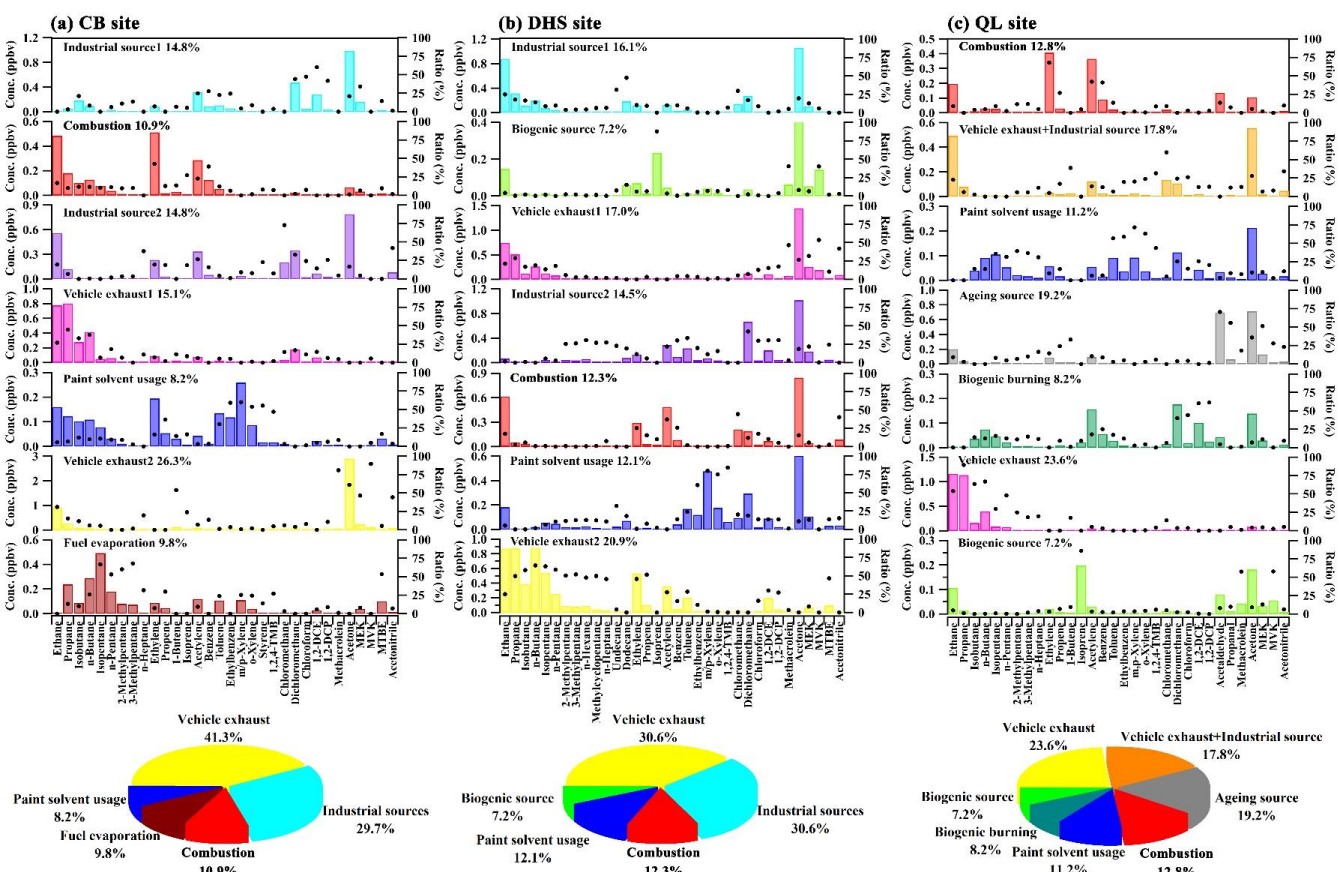

**Figure 8: Source profiles and contributions of VOCs in the CB, DHS, and QL sites during the observation period.**

Note. Bars represent the concentration of each species apportioned to the factor, and black dots represent the percent of each species apportioned to the factor.

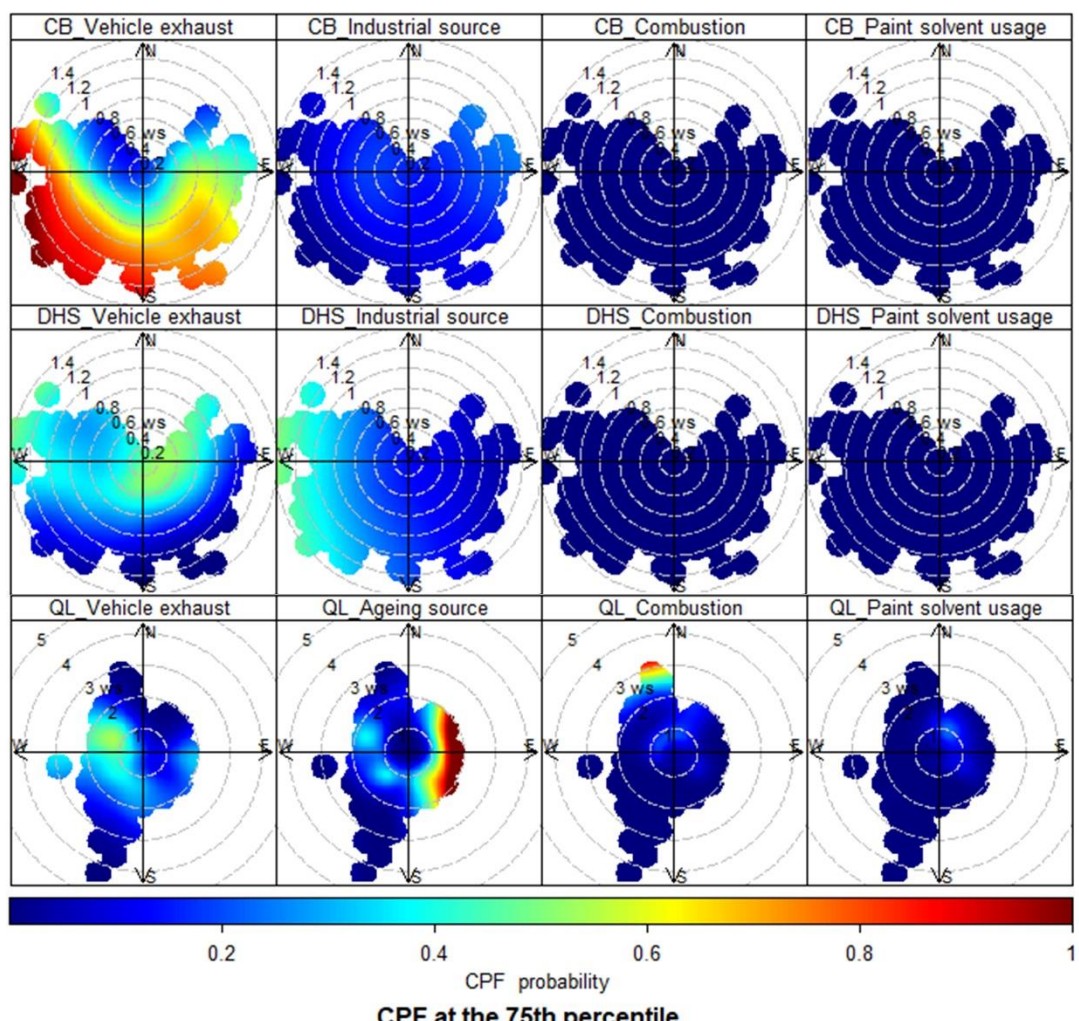

**Figure 9: Polar plot of primary VOC sources in (a) CB, (b) DHS, and (c) QL sites based on the CPF function.**

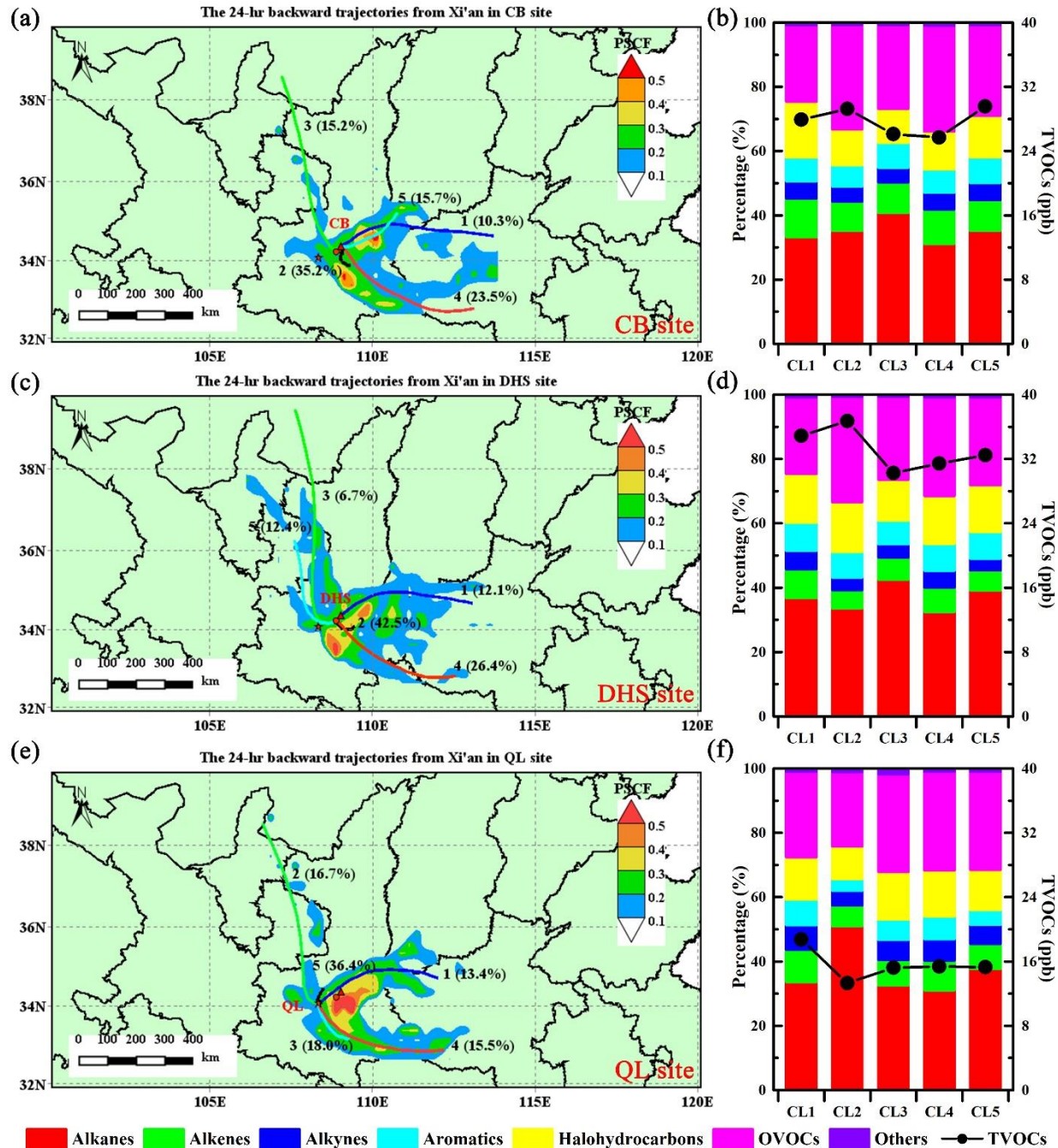

**Figure 10: Backward trajectory cluster analysis (24-h) and PSCF analysis in the CB, DHS, and QL sites.**

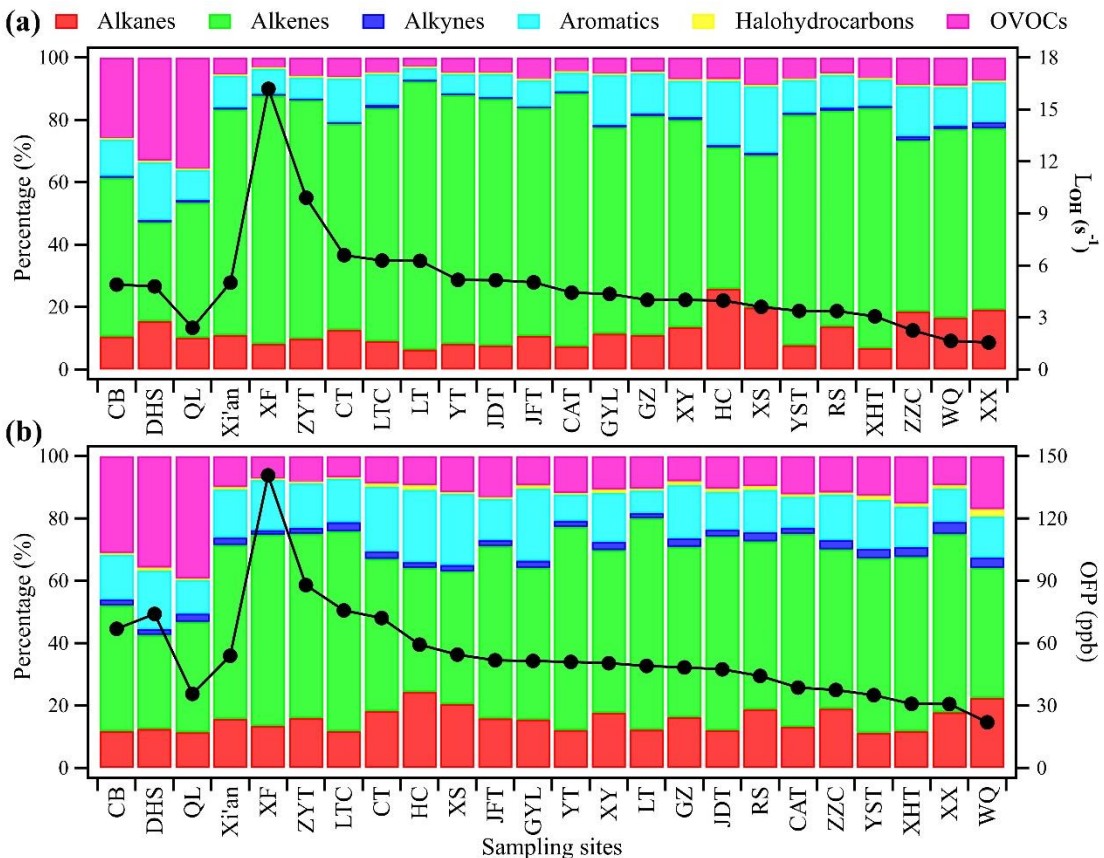


**Figure 11: Contributions of different groups of VOCs to (a) L$_{OH}$ and (b) OFP in different sites in Xi'an.**

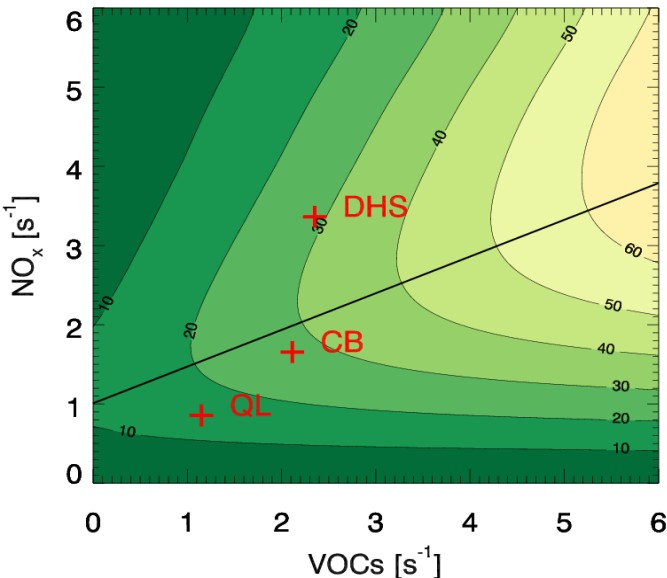

**Figure 12. The ozone production rate (P (O₃)) contours diagram versus anthropogenic VOCs (AVOCs) and NOx using Empirical**
**Kinetic Modelling Approach at CB, DHS, and QL sites.**