# Peer review of "Spatiotemporal Variation, Sources, and Secondary Transformation Potential of VOCs in Xi'an, China"

_Atmospheric Chemistry and Physics, 2020_

## Referee Comment (RC1) · Anonymous Referee #1 · 25 Sep 2020

Song et al. investigated the variation, sources, and chemistry of atmospheric VOCs in Xi'an, China. Field observations were conducted in multiple representative sites in Xi'an. Results showed that vehicle emission was the largest VOC contributor, followed by industrial emissions. Results of backward trajectories coupled with potential source contribution function analysis indicated that Xi'an exhibited a strong local VOC source. In addition, the authors demonstrated that alkenes, aromatics, and OVOCs played dominant roles in the secondary transformation of ambient VOCs in Xi'an. The manuscript is very well written, and the results are clearly presented. Therefore, I would like to recommend its publication in Atmospheric Chemistry and Physics, subject to minor changes.

1. Lines 75-80: GC analysis

[Figure]

Please provide the details of GC procedures (e.g., oven temperature program) for the analyses of both low carbon number and high carbon number compounds.

2. Line 85: "VOC gridded sampling was performed at 7:00 China Standard Time (CST) and 15:00 (CST) on July 1 and July 14, 2019, respectively."

This sentence is confusing. Were the samples collected at both 7:00 and 15:00 on both July 1 and July 14? Or the samples were collected at 7:00 on July 1 and 15:00 on July 14? Please clarify.

3. Line 89: "Ambient air was sampled into a 3-L SiloniteTM 90 canister (Entech Instrument, United States)."

How long was the sampling time? Or what was the time resolution of the VOC sampling? How many samples were collected per site? Please clarify.

4. Lines 116-117: "VOC tracers with a data integrity greater than 75% and greater than 65% valid data (concentration $\geq$ MDL) were selected as the input species."

This sentence is confusing. Do you mean "VOC tracers with greater than 65% valid data"?

Please explain why 75% and 65% were used.

5. Equation (4): please provide references for the uncertainty estimation.

6. Lines 172-176: the effects of temperature

As shown in Figure 2, the temperature during polluted periods could be 10-15 °C higher than that in the clean days. An increase in temperature can enhance both the emissions and the oxidation rates of VOCs.

Can the authors estimate how much of the ozone increase was due to the increase of VOC emissions and how much was due to the enhancement of VOC oxidation rates during the polluted periods?

[Figure]

7. Figure 4: spatial variations of VOCs

Were the TVOC concentrations shown in Figure 4 two-day average values of July 1 and July 14? If no, please specify which date the figure represents. If yes, please explain why the author used the average concentrations. Following comment 2 above, if the sampling was conducted at 7:00 on July 1 and at 15:00 on July 14, were the meteorological conditions similar during the two sampling periods? Would the results be the same by analyzing the data collected on each individual day?

8. Lines 192-194: "In addition, the contribution of OVOCs at the YT site was significantly higher than that of the other sites, indicating that the YT site may be significantly affected by ageing sources (Figure 4a)."

a) Please provide the VOC list identified in this study in the supporting information.

b) What were the OVOC composition measured in this study? Throughout the manuscript, the authors tended to attribute higher OVOC concentration to stronger atmospheric oxidation. Although this is reasonable to some extent, there is a possibility that OVOCs were directly emitted. For example, acetone can be emitted from sources such as solvent evaporation, biomass burning, and vehicle emission, as also shown in Figure 8.

Can the authors comment on the primary emissions of OVOCs during the campaign? And how will this affect the conclusion regarding OVOCs in this study?

9. Section 3.2.1 Specific VOC Ratios

The methodology using VOC ratios to investigate potential sources provides useful insights. However, the uncertainty may be huge. For example, the authors mentioned that in the industrial region, the concentration ratio of toluene to benzene ranged from 3.0 to 6.9, using the results obtained from Zhengzhou city in China (Li et al., 2019a), the Pearl River Delta region (Chan et al., 2006), and several other developed coastal regions in China (Zhang et al., 2015). I would expect that the industry type and composition are likely different between Xi'an and the cities/areas mentioned in the references (e.g., coastal regions). How does the T/B ratio vary from location to location? How will this affect the conclusions of this study?

Similar issues may exist in the T/B ratios for other sources. For example, the T/B ratio for vehicle emissions can be strongly influenced by vehicle type and fuel composition. Please discuss the uncertainty of using these ratios.

10. Figures 6a-6c: what are the green lines?

11. Lines 297-301: "There were two trajectory clusters from the southeast direction, the southeast short distance trajectories (Cluster 2) and southeast medium-long distance trajectories (Cluster 4), accounting for 35.2% and 23.5%, respectively. This result indicated that the VOC concentration in the CB site was significantly affected by the southeast trajectory from the junction of the Shaanxi Province, Hubei Province, and Henan Province in addition to local sources."

The PSCF analysis was based on the results of the 24-h backward trajectories. The lifetimes of different VOCs are different in the atmosphere, as also indicated in Table 2. For example, some OVOC species can have much longer lifetimes than reactive alkenes such as ethylene. The long-lived OVOC species may survive through atmospheric oxidation and get transported to Xi'an over 24 h from surrounding provinces. However, reactive species such as ethylene may not be able to.

Can the authors incorporate the lifetime information of different categories of VOCs into the PSCF analysis?

12. Lines 349-350: "the O3 350 concentration in Xi'an urban areas (CB and DHS) often exceeded the national hourly standard of 200 $\mu$g/m3 (approximately 101.9 ppb)."

In lines 43-48, the author used a different national standard (i.e., 160 $\mu$g/m3). Please be consistent throughout the manuscript.

13. NOx concentration

This study demonstrated that high VOC concentration is a major concern in reducing ozone pollution in Xi'an. However, in addition to VOCs, NOx also plays an indispensable role in tropospheric ozone production. NOx concentration was measured in this study. However, there was little discussion on the effects of NOx concentration on ozone and the interplay among NOx, VOC, and ozone. With the measurement data available, can the authors briefly comment on which regime (VOC-limiting or NOx-limiting) was discussed in this study (e.g., in the urban sites and the rural sites) and corresponding implications?

Technical comments:

1. Line 11: "as a critical precursors of ozone. . .", remove "a"

2. Line 76: change "low-carbon" and "high-carbon" to "low carbon number" and "high carbon number"

3. Line 187: "Of the sites, XF site exhibited the highest VOC concentration of 54 ppb, followed by CT, HC, with concentrations of 41.4, and 38.2 ppb, respectively"

Please keep consistent the number of significant figures throughout the manuscript.

4. Line 199: missing "to" between "used" and "preliminarily"

---

## Referee Comment (RC2) · Anonymous Referee #3 · 1 Dec 2020

General comments:

Understanding the sources of VOCs is vital to the mitigation of O3 pollution. Song et al. performed comprehensive field observations and VOC grid sampling in Xi'an to elucidate the concentration levels, source characteristics, and secondary conversion ability of VOCs. They found that vehicle exhaust was the dominant source of VOC emissions in Xi'an. This paper has important implications for the control of O3 pollution in megacities, so it is well within the scope of ACP. I recommend this paper to be published after addressing the comments below.

Specific comments:

Lines 33-38: The authors only present VOC sources for some specific cities. Can the

authors summarize the results of previous studies for various regions of China, e.g. Beijing-Tianjin-Hebei, Yangtze River Delta, Pearl River Delta?

Lines 78-79: How often were external standards run? Please provide more details.

Lines 88-89: What's the duration for sampling? Did the authors also test blank samples?

Lines 158-159: It is recommended to provide the VOC list in the supporting information.

Lines 172-173: It is not clear to me how good the correlation between O3 and temperature is. What is R2? It seems that the correlation is moderate.

Lines 175-180: The average temperature on polluted days is much higher than that on clean days, which will increase the emission of some VOCs, e.g. isoprene as well as solvent evaporation. As the precursor of MVK and MACR, did the concentrations of isoprene increase on the polluted days?

Based on Figure 3, the increase in the concentration of OVOC on O3 pollution days is largely driven by the increased concentration of acetone. The authors also show in section 3.2.2 that acetone is mainly from vehicle exhaust and industrial sources. Both primary emission and/or secondary transformation may contribute to the increase of OVOC. Can the authors estimate the contribution from primary emissions?

Did the NOx concentration change during the polluted and clean days? Did NOx play a role in increasing the O3 concentration on polluted days?

Line 184: It is recommended to provide the VOC list and the grid sampling data in the supporting information.

Line 190: Did "the overall level" mean "the average concentration"?

Lines 192-194: Are there any specific industrial sources near YT? Again, the contribution of primary emissions to OVOCs should be excluded to draw this conclusion.

Lines 209-210: These numbers are the slopes of the fitting lines, not the correlation coefficients. Please also revise other places accordingly.

Lines 215-217: For the grid sampling, only samples at 7:00 and 15:00 were collected. It is reasonable that vehicle exhaust greatly contributed to the overall VOCs because of the sampling time. The authors should state the weakness of the sampling as a caveat.

Section 3.2.2

MVK is a photochemical product of isoprene. Why are most of the MVK attributed to vehicle exhaust at the CB site?

Figure 1: it is recommended to mark Feiwei plain.

Figures 5 and 7: Linear correlations are shown, not correlation coefficients.

Figure 6: What does the green line represent?

Figure 8: What do the bars and dots represent? Please explain in the caption.

Technical corrections:

Line 10: "a critical precursors"...delete "a".

Line 23: References are missing.

Line 35: "source" should be "sources"

Line 36: "indicates" should be "indicate"

Equation 1, lines 98 and 100: the rate coefficient is typically represented by the lower-case k.

---

## Author Comment (AC1) · 14 Jan 2021

Author's response by Mengdi Song et al. Corresponding to li_xin@pku.edu.cn. We would like to thank the editor and the reviewers for the great efforts and elaborate work on this manuscript. We revised the manuscript by responding to each of the suggestions in the reviews. In our response, the questions of the reviewers are shown in Italic form and the responses in standard form.

Please also note the supplement to this comment:
https://acp.copernicus.org/preprints/acp-2020-704/acp-2020-704-AC1-supplement.pdf
* * *
[Figure]

2020.

**Supplement:**

**Response to the Comments of the Reviewers**

Dear Editor and Reviewers,

We would like to thank you and the reviewers for the great efforts and elaborate work on this manuscript.

We revised the manuscript by responding to each of the suggestions in the reviews. In our response, the questions of the reviewers are shown in *Italic* form and the responses in standard form.

We appreciate your help and time.

Sincerely yours,

Xin Li and Co-authors.

College of Environmental Sciences and Engineering Peking University 100871 Beijing China E-mail: li\_xin@pku.edu.cn Tel: +86-185 1358 6831 ------

Manuscript Number: acp-2020-704.

Manuscript Title: Spatiotemporal Variation, Sources, and Secondary Transformation Potential of VOCs in Xi'an, China.

\_\_\_\_\_

**Response to Reviewer #1**

**General comments**

Song et al. investigated the variation, sources, and chemistry of atmospheric VOCs in Xi'an, China. Field observations were conducted in multiple representative sites in Xi'an. Results showed that vehicle emission was the largest VOC contributor, followed by industrial emissions. Results of backward trajectories coupled with potential source contribution function analysis indicated that Xi'an exhibited a strong local VOC source. In addition, the authors demonstrated that alkenes, aromatics, and OVOCs played dominant roles in the secondary transformation of ambient VOCs in Xi'an. The manuscript is very well written, and the results are clearly presented. Therefore, I would like to recommend its publication in Atmospheric Chemistry and Physics, subject to minor changes.

**Response:**

We would like to thank reviewer #1 for carefully reading our manuscript and for your valuable and constructive comments. We carefully reviewed and improved each part according to the reviewer's suggestions. Listed below are our point-by-point responses to reviewer's comments. Lastly, we would like to thank reviewer for the positive comments again.

**Comments**

1. Lines 75-80: GC analysis.

*Please provide the details of GC procedures (e.g., oven temperature program) for the analyses of both low carbon number and high carbon number compounds.*

**Response:**

We appreciate the reviewer's comments, and we have added more information on the GC analysis. Now it reads as follows:

The GC–MS/FID instrument utilized a dual gas path separation method. The sample gas, after water and CO2 were removed, captured VOC components through an ultra-low temperature system (-160 °C), and the gas chromatography analysis system was utilized after thermal desorption. The oven temperature was programmed at 37 °C maintained for 5 min initially, then raised to 120 °C at 5 °C min-1 holding for 5 min and latter to 180 °C at 6 °C min-1 holding for 5 min. The low carbon number (C2–C5) compounds were separated on an Al2O3/KCl PLOT column and quantified using the FID (200 °C). The high carbon number (C5–C10) compounds were separated on a DB-624 column and quantified using MS (230 °C).

\_\_\_\_\_

2. Line 85: "VOC gridded sampling was performed at 7:00 China Standard Time (CST) and 15:00 (CST) on July 1 and July 14, 2019, respectively."

*This sentence is confusing. Were the samples collected at both 7:00 and 15:00 on both July 1 and July 14? Or the samples were collected at 7:00 on July 1 and 15:00 on July 14? Please clarify.*

**Response:**

We appreciate the reviewer's comments, and we have clarified this sentence in the revised manuscript. Now the sentence reads as follows:

VOC gridded samples were collected at each site for two days (July 1 and July 14, 2019) and twice a day at 7:00 China Standard Time (CST) and 15:00 (CST). A total of 80 ambient air samples with a frequency of 4 samples per site were collected and each sample was stored in a 3.2-L SiloniteTM canister (Entech Instrument, United States).

\_\_\_\_\_

3. Line 89: "Ambient air was sampled into a 3-L SiloniteTM 90 canister (Entech Instrument, United States)."

How long was the sampling time? Or what was the time resolution of the VOC sampling? How many samples were collected per site? Please clarify.

**Response:**

We appreciate the reviewer's comments, and we have added the sampling information in the revised manuscript. Now it reads as follows:

VOC gridded samples were collected at each site for two days (July 1 and July 14, 2019) and twice a day at 7:00 China Standard Time (CST) and 15:00 (CST). The gridded sampling site were chosen based on the technical regulations for selecting ambient air quality monitoring stations (HJ 664-2013) and method for selection of Photochemical Assessment Monitoring Stations (EPA). According to the prevailing wind direction and to ensure coverage of all urban areas in Xi'an, 20 gridded sampling sites were selected for this study (Figure 1). Detailed sampling site information is shown in Table 1. A total of 80 ambient air samples with a frequency of 4 samples per site were collected and each sample was stored in a 3.2-L SiloniteTM canister (Entech Instrument, United States). Before VOC gridded sampling, the SiloniteTM canisters were cleaned with high purity nitrogen using the Entech 3100 canister cleaning system, and then they were evacuated to a vacuum. Instantaneous sampling method was adopted for ambient air sample collection with a sampling duration of approximately 2 min.

\_\_\_\_\_

4. Lines 116-117: "VOC tracers with a data integrity greater than 75% and greater than 65% valid data (concentration  $\geq$  MDL) were selected as the input species."

This sentence is confusing. Do you mean "VOC tracers with greater than 65% valid data"? Please explain why 75% and 65% were used.

**Response:**

We appreciate the reviewer's comments. In order to ensure the accuracy of PMF model simulation, species with more than 25% missing samples or with more than 35% samples below the method detection limit (MDL) should be excluded in input files (Liu et al., 2020). Therefore, VOC tracers with a data integrity greater than 75% and valid data (concentration  $\geq$  MDL) greater than 65% were selected for this study.

We apologize for the unclear presentation. We have carefully revised this statement in the revised manuscript. Now it reads as follows:

VOCs tracers were selected according to the reported typical emission source profiles in China (He et al., 2015; Liu et al., 2008a; Song et al., 2018). Only those tracers which have data coverage greater than 75% during the campaign and have 65% measured concentrations above the MDL were included in

the PMF analysis (c.f. Liu et al., 2020).

5. Equation (4): please provide references for the uncertainty estimation.

**Response:**

We appreciate the reviewer's comments. We have added references for the uncertainty estimation according to the reviewer's suggestion. Now it reads as follows:

For the PMF input, data below the detection limit were assigned with MDL/2, and the missing data were substituted with median concentration. The uncertainty is calculated using Eq. (4) as follows (Brown et al, 2015; Liu et al, 2016a):

| (     | $\sqrt{(\text{Error Fraction} \times \text{Conc.})^2 + (0.5 \times \text{MDL})^2}$ | Conc. > MDL          |    |
|-------|------------------------------------------------------------------------------------|----------------------|----|
| Unc = | $\frac{5}{6}$ × Conc.                                                              | $Conc. \le MDL $ (4) | 4) |
| (     | $4 \times median$ conc.                                                            | Missing data         |    |

**6. Lines 172-176: the effects of temperature**

As shown in Figure 2, the temperature during polluted periods could be 10-15 °C higher than that in the clean days. An increase in temperature can enhance both the emissions and the oxidation rates of VOCs. Can the authors estimate how much of the ozone increase was due to the increase of VOC emissions and how much was due to the enhancement of VOC oxidation rates during the polluted periods?

**Response:**

We appreciate the reviewer's comments. We agree with reviewer that an increase in temperature can enhance the emissions and the oxidation rates of some VOCs (such as isoprene and solvent evaporation). Figure 3 shows that the emission of isoprene increased significantly during the polluted days, along with the increase of temperature (Figure 3c, 3f). However, concentrations of most aromatics which were regarded as tracers of solvent evaporation remained unchanged (Figure 3c, 3f). Therefore, we think there is no clear evidence that the emissions of solvent evaporation increased on the polluted days.

We agree with the reviewer that the elevated concentration of OVOCs on  $O_3$  pollution days could indicate the enhancement of VOC oxidation rates at this stage. However, quantifying the contribution of emission and meteorology (e.g., temperature) to  $O_3$  formation would require detailed emission-based model analysis which is beyond the scope of this manuscript and will be discussed in a separate paper. We have modified the Figure 3 and add the following discussion in the revised manuscript.

Figure 3: Diurnal variations in wind speed (WS), temperature (T), O3, NOx, and TVOCs on clean and polluted days at the (a) and (b) CB and (d) and (e) DHS sites. Differences in VOC concentrations between clean and polluted days at the (c) CB and (f) DHS sites. Note: ALK = alkanes, ALE = alkenes (except isoprene), ISO = isoprene, ALY = alkynes, ARO =

aromatics, HALO = Halohydrocarbons, MVK = Methyl Vinyl Ketone, and MACR = Methacrolein.

We have also modified this part in the revised manuscript. Now it reads as follows:

As shown in Figure 3, isoprene concentrations at urban sites increased significantly during the  $O_3$  pollution day, which could due to the stronger plant emission at elevated temperature (Guenther et al., 1993; Guenther et al., 2012; Stavrakou et al., 2014). Concentrations of isoprene oxidation products (i.e., MVK and MACR) as well as most OVOCs also increased in the same period. However, similar concentrations of anthropogenic VOCs are found

---

## Author Comment (AC2) · 14 Jan 2021

Author's response by Mengdi Song et al. Corresponding to li_xin@pku.edu.cn. We would like to thank reviewer #1 for carefully reading our manuscript and for your valuable and constructive comments. We carefully revised and improved each part according to the reviewer's suggestions. Listed below are our point-by-point responses to reviewer's comments. In our response, the questions of the reviewers are shown in Italic form and the responses in standard form.

Please also note the supplement to this comment:
https://acp.copernicus.org/preprints/acp-2020-704/acp-2020-704-AC2-supplement.pdf

**Supplement:**

**Response to the Comments of the Reviewers**

Dear Editor and Reviewers,

We would like to thank you and the reviewers for the great efforts and elaborate work on this manuscript.

We revised the manuscript by responding to each of the suggestions in the reviews. In our response, the questions of the reviewers are shown in *Italic* form and the responses in standard form.

We appreciate your help and time.

Sincerely yours,

Xin Li and Co-authors.

College of Environmental Sciences and Engineering
Peking University
100871 Beijing China
E-mail: li_xin@pku.edu.cn
Tel: +86-185 1358 6831
* * *
*Manuscript Number: acp-2020-704.*

*Manuscript Title: Spatiotemporal Variation, Sources, and Secondary Transformation Potential of VOCs in Xi'an, China.*
* * *
**Response to Reviewer #1**

*General comments*

*Song et al. investigated the variation, sources, and chemistry of atmospheric VOCs in Xi'an, China. Field observations were conducted in multiple representative sites in Xi'an. Results showed that vehicle emission was the largest VOC contributor, followed by industrial emissions. Results of backward trajectories coupled with potential source contribution function analysis indicated that Xi'an exhibited a strong local VOC source. In addition, the authors demonstrated that alkenes, aromatics, and OVOCs played dominant roles in the secondary transformation of ambient VOCs in Xi'an. The manuscript is very well written, and the results are clearly presented. Therefore, I would like to recommend its publication in Atmospheric Chemistry and Physics, subject to minor changes.*

**Response:**

We would like to thank reviewer #1 for carefully reading our manuscript and for your valuable and constructive comments. We carefully revised and improved each part according to the reviewer's suggestions. Listed below are our point-by-point responses to reviewer's comments. Lastly, we would like to thank reviewer for the positive comments again.
* * *
*Comments*

*1. Lines 75-80: GC analysis.*

*Please provide the details of GC procedures (e.g., oven temperature program) for the analyses of both low carbon number and high carbon number compounds.*

**Response:**

We appreciate the reviewer's comments, and we have added more information on the GC analysis. Now it reads as follows:

The GC–MS/FID instrument utilized a dual gas path separation method. The sample gas, after water and $CO_2$ were removed, captured VOC components through an ultra-low temperature system (-160 °C), and the gas chromatography analysis system was utilized after thermal desorption. The oven temperature was programmed at 37 °C maintained for 5 min initially, then raised to 120 °C at 5 °C $min^{-1}$ holding for 5 min and latter to 180 °C at 6 °C $min^{-1}$ holding for 5 min. The low carbon number ($C_2$–$C_5$) compounds were separated on an $Al_2O_3$/KCl PLOT column and quantified using the FID (200 °C). The high carbon number ($C_5$–$C_{10}$) compounds were separated on a DB-624 column and quantified using MS (230 °C).
* * *
*2. Line 85: "VOC gridded sampling was performed at 7:00 China Standard Time (CST) and 15:00 (CST) on July 1 and July 14, 2019, respectively."*

*This sentence is confusing. Were the samples collected at both 7:00 and 15:00 on both July 1 and July 14? Or the samples were collected at 7:00 on July 1 and 15:00 on July 14? Please clarify.*

**Response:**

We appreciate the reviewer's comments, and we have clarified this sentence in the revised manuscript. Now the sentence reads as follows:

VOC gridded samples were collected at each site for two days (July 1 and July 14, 2019) and twice a day at 7:00 China Standard Time (CST) and 15:00 (CST). A total of 80 ambient air samples with a frequency of 4 samples per site were collected and each sample was stored in a 3.2-L Silonite$^{TM}$ canister (Entech Instrument, United States).
* * *
*3. Line 89: "Ambient air was sampled into a 3-L SiloniteTM 90 canister (Entech Instrument, United States)."*

*How long was the sampling time? Or what was the time resolution of the VOC sampling? How many samples were collected per site? Please clarify.*

**Response:**

We appreciate the reviewer's comments, and we have added the sampling information in the revised manuscript. Now it reads as follows:

VOC gridded samples were collected at each site for two days (July 1 and July 14, 2019) and twice a day at 7:00 China Standard Time (CST) and 15:00 (CST). The gridded sampling site were chosen based on the technical regulations for selecting ambient air quality monitoring stations (HJ 664-2013) and method for selection of Photochemical Assessment Monitoring Stations (EPA). According to the prevailing wind direction and to ensure coverage of all urban areas in Xi'an, 20 gridded sampling sites were selected for this study (Figure 1). Detailed sampling site information is shown in Table 1. A total of 80 ambient air samples with a frequency of 4 samples per site were collected and each sample was stored in a 3.2-L Silonite$^{TM}$ canister (Entech Instrument, United States). Before VOC gridded sampling, the Silonite$^{TM}$ canisters were cleaned with high purity nitrogen using the Entech 3100 canister cleaning system, and then they were evacuated to a vacuum. Instantaneous sampling method was adopted for ambient air sample collection with a sampling duration of approximately 2 min.
* * *
*4. Lines 116-117: "VOC tracers with a data integrity greater than 75% and greater than 65% valid data (concentration ≥ MDL) were selected as the input species."*

*This sentence is confusing. Do you mean "VOC tracers with greater than 65% valid data"?*

*Please explain why 75% and 65% were used.*

**Response:**

We appreciate the reviewer's comments. In order to ensure the accuracy of PMF model simulation, species with more than 25% missing samples or with more than 35% samples below the method detection limit (MDL) should be excluded in input files (Liu et al., 2020). Therefore, VOC tracers with a data integrity greater than 75% and valid data (concentration ≥ MDL) greater than 65% were selected for this study.

We apologize for the unclear presentation. We have carefully revised this statement in the revised manuscript. Now it reads as follows:

VOCs tracers were selected according to the reported typical emission source profiles in China (He et al., 2015; Liu et al., 2008a; Song et al., 2018). Only those tracers which have data coverage greater than 75% during the campaign and have 65% measured concentrations above the MDL were included in

the PMF analysis (c.f. Liu et al., 2020).
* * *
*5. Equation (4): please provide references for the uncertainty estimation.*

**Response:**

We appreciate the reviewer's comments. We have added references for the uncertainty estimation according to the reviewer's suggestion. Now it reads as follows:

For the PMF input, data below the detection limit were assigned with MDL/2, and the missing data were substituted with median concentration. The uncertainty is calculated using Eq. (4) as follows (Brown et al, 2015; Liu et al, 2016a):

$$\text{Unc} = \begin{cases} \sqrt{(\text{Error Fraction} \times \text{Conc.})^2 + (0.5 \times \text{MDL})^2} & \text{Conc.} > \text{MDL} \\ \frac{5}{6} \times \text{Conc.} & \text{Conc.} \leq \text{MDL} \\ 4 \times \text{median conc.} & \text{Missing data} \end{cases} \quad (4)$$
* * *
*6. Lines 172-176: the effects of temperature*

*As shown in Figure 2, the temperature during polluted periods could be 10-15 °C higher than that in the clean days. An increase in temperature can enhance both the emissions and the oxidation rates of VOCs. Can the authors estimate how much of the ozone increase was due to the increase of VOC emissions and how much was due to the enhancement of VOC oxidation rates during the polluted periods?*

**Response:**

We appreciate the reviewer's comments. We agree with reviewer that an increase in temperature can enhance the emissions and the oxidation rates of some VOCs (such as isoprene and solvent evaporation). Figure 3 shows that the emission of isoprene increased significantly during the polluted days, along with the increase of temperature (Figure 3c, 3f). However, concentrations of most aromatics which were regarded as tracers of solvent evaporation remained unchanged (Figure 3c, 3f). Therefore, we think there is no clear evidence that the emissions of solvent evaporation increased on the polluted days.

We agree with the reviewer that the elevated concentration of OVOCs on $O_3$ pollution days could indicate the enhancement of VOC oxidation rates at this stage. However, quantifying the contribution of emission and meteorology (e.g., temperature) to $O_3$ formation would require detailed emission-based model analysis which is beyond the scope of this manuscript and will be discussed in a separate paper.

We have modified the Figure 3 and add the following discussion in the revised manuscript.

[Figure]

**Figure 3: Diurnal variations in wind speed (WS), temperature (T), O₃, NOx, and TVOCs on clean and polluted days at the (a) and (b) CB and (d) and (e) DHS sites. Differences in VOC concentrations between clean and polluted days at the (c) CB and (f) DHS sites.**

Note: ALK = alkanes, ALE = alkenes (except isoprene), ISO = isoprene, ALY = alkynes, ARO = aromatics, HALO = Halohydrocarbons, MVK = Methyl Vinyl Ketone, and MACR = Methacrolein.

We have also modified this part in the revised manuscript. Now it reads as follows:

As shown in Figure 3, isoprene concentrations at urban sites increased significantly during the O₃ pollution day, which could due to the stronger plant emission at elevated temperature (Guenther et al., 1993; Guenther et al., 2012; Stavrakou et al., 2014). Concentrations of isoprene oxidation products (i.e., MVK and MACR) as well as most OVOCs also increased in the same period. However, similar concentrations of anthropogenic VOCs are found in clean and polluted days. This indicates a stronger photochemical conversion of VOCs existed in O₃ pollution days, which could due to the more favorable meteorological conditions (i.e., higher temperature and solar radiation).
* * *
*7. Figure 4: spatial variations of VOCs*

*Were the TVOC concentrations shown in Figure 4 two-day average values of July 1 and July 14? If no, please specify which date the figure represents. If yes, please explain why the author used the average concentrations. Following comment 2 above, if the sampling was conducted at 7:00 on July 1 and at 15:00 on July 14, were the meteorological conditions similar during the two sampling periods? Would the results be the same by analyzing the data collected on each individual day?*

**Response:**

The TVOC concentrations shown in Figure 4 is two-day (4 samples) average values of July 1 and July 14. In this study, a total of 80 ambient air samples with a frequency of 4 samples per site were collected. From the Table S4 we can see the meteorological conditions similar during the two sampling periods. This table has been included in the revised Supplement.

**Table S4. The weather parameter on July 1 and July 14 in Xi'an**

| Date | Area | WS (m/s) | WD (°) | T (°C) | RH (%) |
|------|------|----------|--------|--------|--------|
| 2019/7/1 | urban | 0.6 | 205.4 | 28.9 | 53.3 |
| 2020/7/14 | | 0.6 | 199.7 | 31.3 | 50.9 |
| 2020/7/1 | rural | 1.1 | 209.8 | 27.7 | 54.1 |
| 2020/7/14 | | 1.5 | 201.1 | 29.8 | 50.6 |

Since the weather, emission and transmission conditions of the two days of sampling cannot be completely consistent (R1), in order to reduce the contingency of sampling, we used the method of average concentration to analyze the spatial variation.

[Figure]

**Figure R1: TVOCs concentrations in different sites in Xi'an on July 1 and July 14, 2019.**

We appreciate the reviewer's comments. We have revised the caption in Figure 4 to clarify that the figure indicates the average concentration of TVOC at each sampling site. Now the sentence reads as follows:

Figure 4: (a) Proportions of seven VOCs groups and averaged TVOCs concentrations in different sites in Xi'an. (b) Spatial distribution of averaged TVOC concentrations in different sites in Xi'an.

We have also added the sampling information in the revised manuscript. Now it reads as follows:

VOC gridded samples were collected at each site for two days (July 1 and July 14, 2019) and twice a day at 7:00 China Standard Time (CST) and 15:00 (CST). A total of 80 ambient air samples with a frequency of 4 samples per site were collected and each sample was stored in a 3.2-L Silonite™ canister (Entech Instrument, United States).
* * *
*8. Lines 192-194: "In addition, the contribution of OVOCs at the YT site was significantly higher than that of the other sites, indicating that the YT site may be significantly affected by ageing sources (Figure 4a)."*

*a) Please provide the VOC list identified in this study in the supporting information.*

*b) What were the OVOC composition measured in this study? Throughout the manuscript, the authors tended to attribute higher OVOC concentration to stronger atmospheric oxidation. Although this is reasonable to some extent, there is a possibility that OVOCs were directly emitted. For example, acetone can be emitted from sources such as solvent evaporation, biomass burning, and vehicle emission, as also shown in Figure 8.*

*Can the authors comment on the primary emissions of OVOCs during the campaign?*

*And how will this affect the conclusion regarding OVOCs in this study?*

**Response:**

a) We appreciate the reviewer's comments. We add a list of measured VOCs in the supporting information. The description of VOC measurement in the manuscript is revised as following.

During the field observation campaign, 99 VOCs were measured, including 29 alkanes, 11 alkenes, 1 alkyne, 16 aromatics, 28 halohydrocarbons, 13 oxygenated VOCs (OVOCs), and 1 acetonitrile (Table S1). In the VOC grid sampling, 106 VOCs were measured, including 29 alkanes, 11 alkenes, 1 alkyne, 17 aromatics, 35 halohydrocarbons, 12 OVOCs, and carbon disulfide (Table S2).

b) We appreciate the reviewer's comments. The sources of OVOCs can be divided into anthropogenic primary sources, anthropogenic secondary sources, biogenic sources and background sources (Li et al., 2014; Wang et al., 2015). The multi-linear regression model was used to analyse the sources of OVOCs in different sites in Xi'an. Ethyne, PAN and isoprene were selected as the tracers of the anthropogenic primary source, the anthropogenic secondary source and the biogenic sources respectively. The equation of the multi-linear regression model is as follows:

$$[OVOCs]=k_0+k_1\times[Ethyne]+k_2\times[PAN]+k_3\times[Isoprene] \qquad (R1)$$

where [Ethyne] represents the concentration of Ethyne, [PAN] represents the concentration of PAN, [Isoprene] represents the concentration of isoprene, $k_0$ represents the background concentration, $k_1$, $k_2$ and $k_3$ are the corresponding coefficients.

It can be seen from the figure that the anthropogenic secondary sources are more significant for OVOCs in rural sites. The YT site is a rural site and does not have many primary sources of VOCs (Figure R2). We therefore infer that the source of OVOCs at this site is most likely from aging sources.

In addition, based on the analysis of the multi-linear regression model, we have a deeper understanding of the source of OVOCs during the ozone pollution period. From Figure S1 and S2 we found that the contribution of anthropogenic primary sources to OVOCs on $O_3$ pollution days is more significant.

We have carefully revised this statement in the revised manuscript. Now it reads as follows:

The specialty of OVOCs is that in addition to the primary emissions, OVOCs can also be formed through photochemical oxidation with alkenes and aromatics (Birdsall and Elrod 2011). The sources of OVOCs can be divided into anthropogenic primary sources, anthropogenic secondary sources, biogenic sources and background sources (Li et al., 2014; Wang et al., 2015). Base on the multi-linear regression model results (Figure S1 and S2) we found that the contribution of anthropogenic primary sources to OVOCs on $O_3$ pollution days is more significant.

[Figure]

**Figure S1. Time series of measured OVOCs concentrations and OVOCs calculated from the multi-linear regression model.**

**Note.** The equation of the multi-linear regression model is:

$$[OVOCs]=k_0+k_1\times[Ethyne]+k_2\times[PAN]+k_3\times[Isoprene]$$

where [Ethyne] represents the concentration of Ethyne, [PAN] represents the concentration of PAN, [Isoprene] represents the concentration of isoprene, $k_0$ represents the background concentration, $k_1$, $k_2$ and $k_3$ are the corresponding coefficients, meas. represents measure.

[Figure]

**Figure S2. Contributions of different sources of OVOCs in different sites in Xi'an base on the multi-linear regression model.**

[Figure]

**Figure R2. Geographic environment map of TY site.**
* * *
*9. Section 3.2.1 Specific VOC Ratios*

*The methodology using VOC ratios to investigate potential sources provides useful insights. However, the uncertainty may be huge. For example, the authors mentioned that in the industrial region, the concentration ratio of toluene to benzene ranged from 3.0 to 6.9, using the results obtained from Zhengzhou city in China (Li et al., 2019a), the Pearl River Delta region (Chan et al., 2006), and several other developed coastal regions in China (Zhang et al., 2015). I would expect that the industry type and composition are likely different between Xi'an and the cities/areas mentioned in the references*

*(e.g., coastal regions). How does the T/B ratio vary from location to location? How will this affect the conclusions of this study?*

*Similar issues may exist in the T/B ratios for other sources. For example, the T/B ratio for vehicle emissions can be strongly influenced by vehicle type and fuel composition.*

*Please discuss the uncertainty of using these ratios.*

**Response:**

We appreciate the reviewer's comments. We agree with reviewer that the T/B ratio vary from different industry type and composition. We summarized the T/B ratios of different types of sources in different regions based on a large number of source emission references during recently years (Tsai et al., 2008; Liu et al., 2008a; Yuan et al., 2010; Li et al., 2011; Qiao et al., 2012; Dai et al., 2013; Wang et al., 2013; Yao et al., 2013; Zhang et al., 2013; Zheng et al., 2013; Wang et al., 2014; Mo et al., 2015; Shi et al., 2015; Yao et al., 2015b; Mo et al., 2016; Yao et al., 2015a; Deng et al., 2018), as shown in Table S5.

According to the Report on the China Statistical Yearbook-2020 released by the National Bureau of Statistics of China (http://www.stats.gov.cn/tjsj/ndsj/), Xi'an's industries mainly include petrochemical industry, chemical industry and power plant. As can be seen from the Table S5, the ratio of T/B ranged from $1.4 \pm 0.8$ to $5.8 \pm 3.4$ by different process unit in petrochemical industry, chemical industry and power plant emissions (Mo et al., 2015; Shi et al., 2015). In vehicle source emission researches, the ratio of T/B ranged from $0.9 \pm 0.6$ to $2.2 \pm 0.5$ by different vehicle type and fuel composition (Qiao et al., 2012; Dai et al., 2013; Wang et al., 2013; Yao et al., 2013; Zhang et al., 2013; Yao et al., 2015b; Mo et al., 2016; Yao et al., 2015a; Deng et al., 2018). However, vehicle emissions include both diesel vehicles emissions and gasoline vehicles emissions in the atmospheric environment. Moreover, according to the Report on the China Statistical Yearbook-2020 released by the National Bureau of Statistics of China (http://www.stats.gov.cn/tjsj/ndsj/), there is no obvious difference in the composition of vehicles in various provinces of China (Figure R3). Thus, the ratio of T/B in the traffic source should be closer to

the results of the tunnel experiments which approximately 1.5 ± 0.1 (Liu et al., 2008a; Deng et al., 2018). In addition, the ratio of T/B was greater than 8.8 ± 6.5 by different solvent use process in paint solvent usage source emission researches (Yuan et al., 2010; Wang et al., 2014; Zheng et al., 2013), and the T/B ratio was approximately 0.3 ± 0.1 in different combustion process and raw materials (Liu et al., 2008a; Li et al., 2011; Mo et al., 2016).

[Figure]

**Figure R3. The composition of vehicles in various provinces of China.**

Note. BJ=Beijing, TJ=Tianjin, HB=Hebei, LN=Liaoning, SH=Shanghai, JS=Jiangsu, ZJ=Zhejiang, AH=Anhui, GD=Guangdong, SX=Shanxi. The fuel of large passenger vehicle and heavy-duty vehicles is generally diesel. The fuel of light passenger vehicle is generally gasoline.

Then, we marked the T/B range of industrial sources, vehicle sources, paint solvent usage sources, and burning source in the Figure 5. In addition, in order to reduce the influence of photochemical reaction on the ratio of benzene to toluene, this study selected the weaker photochemical reaction period (3:00-7:00) for the analysis of toluene and benzene (Figure 5). Figure 5 shows that the ratios of toluene to benzene at the CB, DHS, QL, and gridded sampling sites were 1.1 ($R_{Pearson}$=0.5), 3.6 ($R_{Pearson}$=0.6), 0.5 ($R_{Pearson}$=0.8), and 1.75 ($R_{Pearson}$=0.9), respectively. In the urban areas (CB and DHS sites), most of the T/B ratios were distributed within the reference range of vehicle emissions and industrial emissions (Figure 5a, 5b), implying that vehicle sources and industrial sources contribute significantly to the VOCs in Xi'an urban area. In addition, the T/B value of some samples is greater than 5.8 in urban area which may affected by paint solvent usage source (Figure 5b). However, the detailed source contribution needs to be obtained through PMF source analysis results (Section 3.2.2). In the rural area (QL site), most of the T/B ratios were distributed within the reference range of vehicle emissions and burning emissions (Figure 5c), implying that vehicle sources and burning sources contribute significantly to the VOCs in Xi'an rural area. In the gridded sampling sites, the T/B ratio was predominately concentrated around 1.5, indicating that vehicle exhaust sources may greatly contributed to the overall VOCs in Xi'an (Figure 5d).

**Table S5. Toluene to benzene ratio (T/B) of different source profiles in different researches (unit: ppb/ppb).**

| Category | Sub Category | Location | T/B | Reference | Sub Category avgerage | Min | Max |
|---|---|---|---|---|---|---|---|
| **Transportation** | Gasoline vehicle exhaust | Tianjin | 1.08 | Wang et al.,2013; Dai et al.,2013 | 1.08 | 0.93 | 2.21 |
| | Motorcycle exhaust | Taiwan | 1.46 | Yao et al,2013 | 1.46 | | |
| | Diesel vehicle exhaust | Beijing | 1.77 | Yao et al,2015a | 0.93 | | |
| | | Beijing | 1.04 | Yao et al,2015a | | | |
| | | Beijing | 1.05 | Yao et al,2015a | | | |
| | | Beijing | 1.28 | Yao et al,2015a | | | |
| | | Xiamen | 0.21 | Mo, et al.,2016 | | | |
| | | Xiamen | 0.22 | Mo, et al.,2016 | | | |
| | LPG vehicle exhaust | Shanghai | 1.05 | Qiao et al..,2012 | 1.05 | | |
| | Rural vehicle exhaust | Beijing | 2.21 | Yao et al.,2015b | 1.64 | | |
| | | Beijing | 1.07 | Yao et al.,2015b | | | |
| | Fuel evaporation | Guangzhou | 1.71 | Zhang et al.,2013 | 2.21 | | |
| | | Guangzhou | 2.71 | Zhang et al.,2013 | | | |
| | Tunnel | Hefei | 1.52 | Deng et al., 2018 | 1.48 | | |
| | | Hefei | 1.48 | Deng et al., 2018 | | | |
| | | Hefei | 1.31 | Deng et al., 2018 | | | |
| | | Hefei | 1.56 | Deng et al., 2018 | | | |
| | | PRD | 1.52 | Liu et al., 2008a | | | |
| **Burning** | Coaling burning | Beijing | 0.24 | Mo, et al.,2016 | 0.24 | 0.23 | 0.38 |
| | Coaling burning | Beijing | 0.38 | Liu et al.,2008a | 0.38 | | |

| | | | | | | | |
|---|---|---|---|---|---|---|---|
| | Wheat | Beijing | 0.23 | Mo, et al.,2016 | 0.23 | | |
| | Maize | Beijing | 0.30 | Mo, et al.,2016 | 0.30 | | |
| | Wood | Beijing | 0.27 | Li et al., 2011 | 0.27 | | |
| **Solvent use** | Architecture paint | Shanghai | 15.34 | Wang et al.,2014 | 8.82 | 8.82 | 51.43 |
| | | Beijing | 2.30 | Yuan et al.,2010 | | | |
| | Furniture paint | Beijing | 32.44 | Yuan et al.,2010 | 31.35 | | |
| | | Shanghai | 42.13 | Wang et al.,2014 | | | |
| | | PRD | 15.08 | Zheng et al.,2013 | | | |
| | | PRD | 34.68 | Zheng et al.,2013 | | | |
| | | Beijing | 32.44 | Yuan et al.,2010 | | | |
| | Surface paint | PRD | 8.17 | Zheng et al.,2013 | 12.63 | | |
| | | PRD | 17.09 | Zheng et al.,2013 | | | |
| | Paint manufacturing | PRD | 6.10 | Zheng et al.,2013 | 15.94 | | |
| | | PRD | 25.78 | Zheng et al.,2013 | | | |
| | Shoemaking | PRD | 21.28 | Zheng et al.,2013 | 25.38 | | |
| | | PRD | 29.47 | Zheng et al.,2013 | | | |
| | Printing | PRD | 38.15 | Zheng et al.,2013 | 51.43 | | |
| | | PRD | 64.71 | Zheng et al.,2013 | | | |
| **Industrial processes** | Petrochemical industry | YRD | 0.95 | Mo, et al.,2015 | 1.37 | 1.37 | 5.76 |
| | | YRD | 2.20 | Mo, et al.,2015 | | | |
| | | YRD | 3.20 | Mo, et al.,2015 | | | |
| | | YRD | 0.42 | Mo, et al.,2015 | | | |
| | | YRD | 0.95 | Mo, et al.,2015 | | | |
| | | YRD | 1.88 | Mo, et al.,2015 | | | |

| | | | | |
|---|---|---|---|---|
| | YRD | 0.39 | Mo, et al.,2015 | |
| | YRD | 0.98 | Mo, et al.,2015 | |
| | YRD | 1.80 | Mo, et al.,2015 | |
| | YRD | 1.98 | Mo, et al.,2015 | |
| | YRD | 1.13 | Mo, et al.,2015 | |
| | YRD | 0.75 | Mo, et al.,2015 | |
| | YRD | 1.21 | Mo, et al.,2015 | |
| Chemical industry | YRD | 2.37 | Mo, et al.,2015 | 5.76 |
| | YRD | 9.14 | Mo, et al.,2015 | |
| Power plant | Liaoning | 2.89 | Shi et a..,2015 | 3.39 |
| | Liaoning | 5.29 | Shi et a..,2015 | |
| | Liaoning | 0.48 | Shi et a..,2015 | |
| | Liaoning | 4.89 | Shi et a..,2015 | |

Note. PRD = Pearl River Delta, YRD=Yangtze River delta.

[Figure]

**Figure 5: Linear correlations between toluene and benzene at the CB, DHS, QL, and gridded sampling sites between 3:00-7:00 during the observation period.**

We carefully revised this section in the revised manuscript. Now it reads as follows:

The T/B ratio is clearly different for various source profiles (Table S5). In industrial source emission researches, the ratio of T/B ranged from 1.4 ± 0.8 to 5.8± 3.4 by different industry type and process unit (Mo et al., 2015; Shi et al., 2015). In traffic source emission researches, the ratio of T/B ranged from 0.9 ± 0.6 to 2.2± 0.5 by different vehicle type and fuel composition (Qiao et al., 2012; Dai et al., 2013; Wang et al., 2013; Yao et al., 2013; Zhang et al., 2013; Yao et al., 2015b; Mo et al., 2016; Yao et al., 2015a; Deng et al., 2018). However, vehicle emissions include both diesel vehicles emissions and gasoline vehicles emissions in the atmospheric environment. Thus, the ratio of T/B in the traffic source should be closer to the results of the tunnel experiments which approximately 1.5 ± 0.1 (Liu et al., 2008a; Deng et al., 2018). In paint solvent usage source emission researches, the ratio of T/B was greater than 8.8 ± 6.5 by different solvent use process (Yuan et al., 2010; Wang et al., 2014; Zheng et al., 2013). In burning source emission researches, the T/B ratio was approximately 0.3 ± 0.1 in different combustion process and raw materials (Liu et al., 2008a; Li et al., 2011; Mo et al., 2016). In order to reduce the influence of photochemical reaction on the ratio of benzene to toluene, this study selected the weaker photochemical reaction period (3:00-7:00) for the analysis of toluene and benzene (Figure 5). Figure 5 shows that the ratios of toluene to benzene at the CB, DHS, QL, and gridded sampling sites were 1.1 ($R_{Pearson}$=0.5), 3.6

($R_{Pearson}$=0.6), 0.5 ($R_{Pearson}$=0.8), and 1.75 ($R_{Pearson}$=0.9), respectively. In the urban areas (CB and DHS sites), most of the T/B ratios were distributed within the reference range of vehicle emissions and industrial emissions (Figure 5a, 5b), implying that vehicle sources and industrial sources contribute significantly to the VOCs in Xi'an urban area. In addition, the T/B value of some samples is greater than 5.8 in urban area which may affected by paint solvent usage source (Figure 5b). However, the detailed source contribution needs to be obtained through PMF source analysis results (Section 3.2.2). In the rural area (QL site), most of the T/B ratios were distributed within the reference range of vehicle emissions and burning emissions (Figure 5c), implying that vehicle sources and burning sources contribute significantly to the VOCs in Xi'an rural area. In the gridded sampling sites, the T/B ratio was predominately concentrated around 1.5, indicating that vehicle exhaust sources may greatly contributed to the overall VOCs in Xi'an (Figure 5d).
* * *
*10. Figures 6a-6c: what are the green lines?*

**Response:**

We appreciate the reviewer's comments. The green line represents the initial emission ratio of m/p-xylene and ethylbenzene, which can be replaced by the highest concentration ratio in periods where the photochemical reaction is weak.

We have added the green line description in the caption in the revised manuscript. Now it reads as follows:

[Figure]

**Figure 6: Diurnal variations in m/p-xylene to ethylbenzene and OH exposure at the CB, DHS, QL, and gridded sampling sites.**

Note: Time is expressed in CST. The green line represents the initial emission ratio of m/p-xylene and ethylbenzene.
* * *
*11. Lines 297-301: "There were two trajectory clusters from the southeast direction, the southeast short distance trajectories (Cluster 2) and southeast medium-long distance trajectories (Cluster 4), accounting for 35.2% and 23.5%, respectively. This result indicated that the VOC concentration in the CB site was significantly affected by the southeast trajectory from the junction of the Shaanxi Province, Hubei*

*Province, and Henan Province in addition to local sources."*

*The PSCF analysis was based on the results of the 24-h backward trajectories. The lifetimes of different VOCs are different in the atmosphere, as also indicated in Table2. For example, some OVOC species can have much longer lifetimes than reactive alkenes such as ethylene. The long-lived OVOC species may survive through atmospheric oxidation and get transported to Xi'an over 24 h from surrounding provinces.*

*However, reactive species such as ethylene may not be able to.*

*Can the authors incorporate the lifetime information of different categories of VOCs into the PSCF analysis?*

**Response:**

We appreciate the reviewer's comments. Because of the chemical lifetime of most VOCs in the atmosphere is much shorter than that of $PM_{2.5}$. Thus, the air mass tracing time should be considered when these methods are applied to the study of VOC contaminants. After compare the chemical lifetime of alkanes (32-253 h), alkenes (3-10 h), aromatics and others (12-228 h) we used a shorter air mass tracing time of 24 hour to analysis the PSCF of TVOCs in this study (Cai et al., 2010). However, when PSCF analysis is performed on VOCs species with different chemical lifetime, the air mass tracking time should be different. We agree with reviewer and have added the lifetime information of different tracer VOCs species into the PSCF analysis in the revised manuscript. Now it reads as follows:

Based on the PSCF analysis for Xi'an (Figures 10a, 10c, and 10e), in the urban sites, high PSCF values were mainly observed to the east and south of Xi'an, and in rural sites, the high PSCF values were primarily observed to the east of Xi'an. Different air mass tracking time (6h, 12h, 24h) were used in the PSCF analysis of different VOC species (Figure S3). Strong chemically active species (e.g., ethylene and xylene) had shorter air mass tracks, with high PSCF values appeared in areas near sites. However, the high PSCF values of long lifetime species (acetone) were found not only near the site but also in the eastern and southern regions of the site. The highest PSCF values of TVOCs appeared in areas near the CB, DHS, and QL sites, which indicated that Xi'an has a strong local source.

[Figure]

**Figure S3: (a)-(c) 6-h backward trajectory PSCF analysis of ethylene in the CB, DHS, and QL sites, (d)-(f) 12-h backward trajectory PSCF analysis of m/p-Xylene in the CB, DHS, and QL sites, (g)-(i) 24-h backward trajectory PSCF analysis of acetone in the CB, DHS, and QL sites.**
* * *
*12. Lines 349-350: "the O3 350 concentration in Xi'an urban areas (CB and DHS) often exceeded the national hourly standard of 200 μg/m3 (approximately 101.9 ppb)."*

*In lines 43-48, the author used a different national standard (i.e., 160 μg/m3). Please be consistent throughout the manuscript.*

**Response:**

We have used the same national standard of $O_3$ in the manuscript. According the Technical Regulation on Ambient Air Quality Index released by the Ministry of Ecology and Environment of the People's Republic of China (http://www.mee.gov.cn/ywgz/fgbz/bz/bzwb/dqhjbh/dqhjzlbz/201203/t20120302_224165.shtml), the annual evaluation standard of $O_3$ is 160 μg/m$^3$ and the hourly evaluation standard of $O_3$ is 200 μg/m$^3$.
* * *
*13. NOx concentration*

*This study demonstrated that high VOC concentration is a major concern in reducing ozone pollution in Xi'an. However, in addition to VOCs, NOx also plays an indispensable role in tropospheric ozone*

*production. NOx concentration was measured in this study. However, there was little discussion on the effects of NOx concentration on ozone and the interplay among NOx, VOC, and ozone. With the measurement data available, can the authors briefly comment on which regime (VOC-limiting or NOx limiting) was discussed in this study (e.g., in the urban sites and the rural sites) and corresponding implications?*

**Response:**

We appreciate the reviewer's comments. We have added the discussion on the interplay among NOx, VOC, and ozone in the section 3.4.2 of the revised manuscript. Now it reads as follows:

**2.6 Empirical Kinetic Modelling Approach**

The traditional empirical kinetic modelling approach (EKMA) is a model sensitivity tests of observation based box model and often used to evaluate the photochemical nonlinear relationship between ozone and precursors NOx and VOCs. The box model is based on the Regional Atmospheric Chemical Mechanisms version 2 (Goliff et al., 2013) updated with Leuven Isoprene Mechanism (Peeters et al., 2009). The definition and mechanism of the observation based box model is described elsewhere in detail (Tan et al., 2017; Tan et al., 2018).

The EKMA curve can be used as a theoretical basis for designing emission reduction strategies to obtain the best ozone pollution reduction method (Jiang et al. 2018, Tan et al. 2018). The model input parameters include temperature, pressure, humidity, photolysis rate constant, $NO_2$, VOCs, etc. In this model, the ozone production rate P ($O_3$) is calculated by the ozone formation rate F ($O_3$) minus the ozone loss rate D ($O_3$), as shown in Eq. (7). The ozone formation rate F ($O_3$) and the ozone loss rate D ($O_3$) can be calculated using Eq. (8) and (9) as follows:

$$P_{O_3} = F_{O_3} - D_{O_3} \tag{7}$$

$$F_{O_3} = k_{HO_2+NO}[HO_2][NO] + k_{(RO_2+NO)_{eff}}[RO_2][NO] \tag{8}$$

$$D_{O_3} = [O^1D][H_2O] + k_{O_3+OH}[O_3][OH] + k_{O_3+HO_2}[O_3][HO_2] + k_{O_3+alkenes}[O_3][alkenes]$$
$$+ k_{OH+NO_2}[OH][NO_2] \tag{9}$$

where $k_{HO_2+NO}$ is represent rate coefficients for the reaction of the NO with $HO_2$ radical ($8.5 \times 10^{-12}$ molecule$^{-1}$·cm$^3$·s$^{-1}$, 298K); $k_{(RO_2+NO)_{eff}}$ is represent effective rate coefficients for the reaction of the NO with $RO_2$ radical ($8.5 \times 10^{-12}$ molecule$^{-1}$·cm$^3$·s$^{-1}$, 298K); $k_{O_3+OH}$ is represent rate coefficients for the reaction of the $O_3$ with OH radical ($7.3 \times 10^{-14}$ molecule$^{-1}$·cm$^3$·s$^{-1}$, 298K); $k_{O_3+HO_2}$ is represent rate coefficients for the reaction of the $O_3$ with $HO_2$ radical ($1.9 \times 10^{-15}$ molecule$^{-1}$·cm$^3$·s$^{-1}$, 298K); $k_{O_3+alkenes}$ is represent rate coefficients for the reaction of the $O_3$ with alkenes ($2.0 \times 10^{-17}$ molecule$^{-1}$·cm$^3$·s$^{-1}$, 298K); $k_{OH+NO_2}$ is represent rate coefficients for the reaction of the $NO_2$ with OH radical ($1.1 \times 10^{-11}$ molecule$^{-1}$·cm$^3$·s$^{-1}$, 298K).

In this study, the average parameters of the entire observation period were used as the input parameters of the model to calculate the ozone concentration in the baseline scenario. Afterwards, the activity change array of VOCs and $NO_X$ were generated by changing each parameter in equal distance steps. The P ($O_3$) contours under these different VOCs and $NO_X$ reactivity conditions are called EKMA curves.

**3.4.2 VOCs-NOx-O$_3$ Sensitivity**

The relationship between the ozone production rates (P (O₃)), anthropogenic VOCs (AVOCs) reactivity and NOx reactivity of the CB, DHS, and QL sites during the observation period was shown in Figure 12. The black curve in the Figure 12 represents the P (O₃) contour, and the black straight line represents the connection line of the P (O₃) turning point (ridgeline), whose slope represents the photochemical parameter $k_{NOx}/k_{AVOCs}$ (Jiang et al., 2018). When the site's $k_{NOx}/k_{AVOCs}$ value is located above the ridgeline, it means that ozone formation is under VOCs-limited regime, otherwise it means that ozone formation is under NOx-limited regime. It can be seen from Figure 12 that the ozone generation of QL site is located in the NOx-limited regime, and reducing NOx can effectively control ozone generation. The ozone generation of DHS site is located in the VOCs-limited regime, and reducing VOCs can effectively control ozone generation. However, CB site is located in the transition regime between VOC- and NOx-limited regimes. Therefore, simultaneous reduction of VOCs and NOx concentration should be considered at CB site to achieve the purpose of controlling O₃.

[Figure]

**Figure 12. The ozone production rate (P (O₃)) contours diagram versus anthropogenic VOCs (AVOCs) and NOx using Empirical Kinetic Modelling Approach at CB, DHS, and QL sites.**

We also have added this part of the results in the conclusion section. Now it reads as follows:
The VOCs-NOx-O₃ sensitivity analysis results showed that the ozone generation of DHS site is located in the VOCs-limited regime, CB in the transition regime between VOC- and NOx-limited regimes, and QL sites is located in the NOx-limited regime. Therefore, reducing VOCs concentration at DHS site, reducing VOCs and NOx concentration at CB site, and reducing NOx concentration at QL site can effectively control ozone generation.
* * *
*Technical comments*
*1. Line 11: "as a critical precursors of ozone…", remove "a"*

**Response:**
We appreciate the reviewer's comments. We are sorry for our mistakes, and we have removed "a" in this sentence.
* * *
*2. Line 76: change "low-carbon" and "high-carbon" to "low carbon number" and "high carbon number"*

**Response:**

We appreciate the reviewer's comments. We have change "low-carbon" and "high-carbon" to "low carbon number" and "high carbon number" in the revised manuscript. Now it reads as follows:

The low carbon number ($C_2$–$C_5$) compounds were separated on an $Al_2O_3$/KCl PLOT column and quantified using the FID. The high carbon number ($C_5$–$C_{10}$) compounds were separated on a DB-624 column and quantified using MS.
* * *
*3. Line 187: "Of the sites, XF site exhibited the highest VOC concentration of 54 ppb, followed by CT, HC, with concentrations of 41.4, and 38.2 ppb, respectively"*
*Please keep consistent the number of significant figures throughout the manuscript.*

**Response:**

We appreciate the reviewer's comments. We have keep consistent the number of significant figures throughout the manuscript. Now it reads as follows:

Of the sites, XF site exhibited the highest VOC concentration of 54.0 ppb, followed by CT, HC, with concentrations of 41.4, and 38.2 ppb, respectively.
* * *
*4. Line 199: missing "to" between "used" and "preliminarily"*

**Response:**

We appreciate the reviewer's comments. We are sorry for our mistakes, and we have added "to" between "used" and "preliminarily" in this sentence.
* * *

---

## Author Comment (AC3) · 14 Jan 2021

Author's response by Mengdi Song et al. Corresponding to li_xin@pku.edu.cn. We would like to thank reviewer #2 for carefully reading our manuscript and for the valuable and constructive comments. The manuscript was significantly revised according to the reviewer's suggestions. Listed below are our point-by-point responses to reviewer's comments. In our response, the questions of the reviewers are shown in Italic form and the responses in standard form.

Please also note the supplement to this comment:
https://acp.copernicus.org/preprints/acp-2020-704/acp-2020-704-AC3-supplement.pdf

[Figure]

[Figure]

**Supplement:**

**Response to the Comments of the Reviewers**

Dear Editor and Reviewers,

We would like to thank you and the reviewers for the great efforts and elaborate work on this manuscript.

We revised the manuscript by responding to each of the suggestions in the reviews. In our response, the questions of the reviewers are shown in *Italic* form and the responses in standard form.

We appreciate your help and time.

Sincerely yours,

Xin Li and Co-authors.

College of Environmental Sciences and Engineering
Peking University
100871 Beijing China
E-mail: li_xin@pku.edu.cn
Tel: +86-185 1358 6831
* * *
Manuscript Number: acp-2020-704.

Manuscript Title: Spatiotemporal Variation, Sources, and Secondary Transformation Potential of VOCs in Xi'an, China.
* * *
**Response to Reviewer #2**

*General comments*

*Understanding the sources of VOCs is vital to the mitigation of O₃ pollution. Song et al. performed comprehensive field observations and VOC grid sampling in Xi'an to elucidate the concentration levels, source characteristics, and secondary conversion ability of VOCs. They found that vehicle exhaust was the dominant source of VOC emissions in Xi'an. This paper has important implications for the control of O₃ pollution in megacities, so it is well within the scope of ACP. I recommend this paper to be published after addressing the comments below.*

**Response:**

We would like to thank reviewer #2 for carefully reading our manuscript and for the valuable and constructive comments. The manuscript was significantly revised according to the reviewer's suggestions. Listed below are our point-by-point responses to reviewer's comments. Lastly, we would like to thank you for your comments and guidance.
* * *
*Comments*

*1. Lines 33-38: The authors only present VOC sources for some specific cities. Can the authors summarize the results of previous studies for various regions of China, e.g. Beijing-Tianjin-Hebei, Yangtze River Delta, Pearl River Delta?*

**Response:**

We appreciate the reviewer's comments. After investigating a large amount of literature, we found that different studies have very different PMF source resolution methods, including source numbers and source definition. We categorize the sources of VOCs in these studies into 7 sources, as shown in Figure R4. From the Figure R4 we found that the concentrations of VOCs in urban areas were seriously affected by vehicle exhaust. In addition, southern China was significantly affected by paint solvent usage sources. However, it is difficult to summarize the results for different regions of China.

[Figure]

**Figure R4. Newly researches about Source apportionment of VOCs at 16 sites in China**

Note: The 16 sites include Beijing (Li et al., 2020), Tianjin (Yang et al., 2019), Langfang (Zhang et al., 2019a), Yuncheng (Gao et al., 2020), Zhengzhou (Zhang et al., 2019b), Xi'an (This study), Wuhan (Shen et al., 2020), Lanzhou (Zhou et al., 2019), Nanjing (An et al., 2017), Guangzhou (Ling et al., 2011), Xiamen (Zhuang et al., 2019), Shanghai (Wang et al., 2013), Chengdu (Song et al., 2018), Chongqing (Li et al., 2018), Taiwan (Chen et al., 2019), Hong Kong (Ling and Guo, 2014).
* * *
*2. Lines 78-79: How often were external standards run? Please provide more details.*

**Response:**

We appreciate the reviewer's comments. We agree with reviewer and carefully revised the manuscript according to the reviewer's suggestion. Now it reads as follows:

External and internal standard gas produced by The Linde Group in the United States were used to calibrate the GC–MS/FID. External standard gas were used to calibrate the GC–MS/FID weekly during the campaign to ensure quantitative accuracy. In addition, the instruments were also daily calibrated by internal standard gases (Bromochloromethane, 1,4-Dichlorobenzene, Chlorobenzene, and Fluorobromobenzene) to ensure the stability of the instrument.
* * *
*3. Lines 88-89: What's the duration for sampling? Did the authors also test blank samples?*

**Response:**

This study uses instantaneous sampling to collect air samples, and the sampling time is about 2 min. In addition, blank sample tests were performed on the VOC gridded sampling in each sampling period (July 1 7:00, July 1 15:00, July 14 7:00, and July 14 15:00). The concentration of all VOCs species in the blank sample is below the method detection limits (MDLs), indicating that the canisters were not contaminated during transportation.

We appreciate the reviewer's comments. We have added the above sampling information in the revised manuscript. Now it reads as follows:

Before VOC gridded sampling, the Silonite™ canisters were cleaned with high purity nitrogen using the Entech 3100 canister cleaning system, and then they were evacuated to a vacuum. Instantaneous sampling method was adopted for ambient air sample collection with a sampling duration of approximately 2min. VOCs in the sampled air were analyzed using a GC–MS/FID system, which was the same as that

used for online measurements, but it was running in off-line mode. In this study, blank sample tests were performed on the VOC gridded sampling in each sampling period (July 1 7:00, July 1 15:00, July 14 7:00, and July 14 15:00). The concentration of all VOCs species in the blank sample is below the MDLs, indicating that the canisters were not contaminated during transportation.
* * *
*4. Lines 158-159: It is recommended to provide the VOC list in the supporting information.*

**Response:**

We appreciate the reviewer's comments. We agree with reviewer and added VOC list in the supporting information. Now it reads as follows:

During the field observation campaign, 99 VOCs were measured, including 29 alkanes, 11 alkenes, 1 alkyne, 16 aromatics, 28 halohydrocarbons, 13 oxygenated VOCs (OVOCs), and 1 acetonitrile (Table S1).

**Table S1: Measured VOC species in CB, DHS, and QL sites.**

| Classification | VOC Species | | |
|---|---|---|---|
| Alkanes | Ethane | 3-Methylpentane | Methylcyclohexane |
| | Propane | n-Hexane | 2,3,4-Trimethylpentane |
| | Iso-butane | 2,4-Dimethylpentane | 2-Methylheptane |
| | n-Butane | Methylcyclopentane | 3-Methylheptane |
| | Cyclopentane | 2-Methylhexane | n-Octane |
| | Iso-pentane | Cyclohexane | n-Nonane |
| | n-Pentane | 2,3-Dimethylpentane | n-Decane |
| | 2,2-Dimethylbutane | 3-Methylhexane | n-Undecane |
| | 2,3-Dimethylbutane | 2,2,4-Trimethylpentane | n-Dodecane |
| | 2-Methylpentane | n-Heptane | |
| Alkenes | Ethene | Cis-butene | Isoprene |
| | propene | 1,3-Butadiene | cis-2-Pentene |
| | Trans-2-butene | 1-Pentene | 1-Hexene |
| | 1-Butene | trans-2-Pentene | |
| Alkynes | Ethyne | | |
| Aromatics | Benzene | iso-Propylbenzene | 1,2,4-Trimethylbenzene |
| | Toluene | n-Propylbenzene | 1,2,3-Trimethylbenzene |
| | Ethylbenzene | m-ethyltoluene | m-diethylbenzene |
| | m/p-Xylene | p-ethyltoluene | p-diethylbenzene |
| | o-Xylene | 1,3,5-Trimethylbenzene | |
| | Styrene | o-Ethyltoluene | |
| Halohydro-carbons | Freon114 | cis-1,2-Dichloroethylene | 1,1,2-trichloroethane |
| | Chloromethane | Chloroform | Tetrachloroethene |
| | Vinylchloride | 1,1,1-Trichloroethane | 1,2-Dibromoethane |
| | Bromomethane | Tetrachloromethane | Chlorobenzene |
| | Chloroethane | 1,2-Dichloroethane | 1,3-Dichlorobenzene |
| | Freon11 | Trichloroethylene | 1,4-Dichlorobenzene |
| | 1,1-Dichloroethene | 1,2-Dichloropropane | Benzylchloride |

| | | | |
|---|---|---|---|
| | Freon113 | Bromodichloromethane | 1,2-Dichlorobenzene |
| | Dichloromethane | trans-1,3-Dichloropropene | |
| | 1,1-Dichloroethane | cis-1,3-Dichloropropene | |
| OVOCs | Acetaldehyde | Methyl Vinyl Ketone | 3-Pentanone |
| | Acrolein | Methyl Ethyl Ketone | n-Hexanal |
| | Propanal | n-Butanal | MTBE |
| | Acetone | 2-Pentanone | |
| | Methacrolein | n-Pentanal | |
| Others | Acetonitrile | | |
* * *
*5. Lines 172-173: It is not clear to me how good the correlation between O3 and temperature is. What is R2? It seems that the correlation is moderate.*

**Response:**

We appreciate the reviewer's comments. From figure R5 we can see, the linear correlations between $O_3$ and temperature on polluted days ($R_{Pearson}=0.7$, $R^2=0.5$) is stronger than that on clean days ($R_{Pearson}=0.5$, $R^2=0.2$) at CB site. The correlation between $O_3$ and temperature at the DHS site shows similar characteristics, and the linear correlations ($R^2$) between $O_3$ and temperature on polluted days and clean days were 0.4 ($R_{Pearson}=0.7$) and 0.3 ($R_{Pearson}=0.5$), respectively.

[Figure]

**Figure R5: Linear correlations between O₃ and temperature on clean and polluted days.**

We also agree with the reviewer that the statement was not very it may not be very appropriate to express a significant positive correlation between $O_3$ and temperature. We carefully revised the statement according to the reviewer's suggestion. Now it reads as follows:

The variation trends of $O_3$ and temperature display a positive correlation, and the linear correlations between $O_3$ and temperature on polluted days ($R_{Pearson}=0.7$) is stronger than that on clean days ($R_{Pearson}=0.5$). The value of temperature, $O_3$, and TVOCs all increased significantly on polluted days, indicating that the secondary transformation of VOCs to $O_3$ is more conducive at high temperatures.
* * *
*6. Lines 175-180: The average temperature on polluted days is much higher than that on clean days, which will increase the emission of some VOCs, e.g. isoprene as well as solvent evaporation. As the precursor of MVK and MACR, did the concentrations of isoprene increase on the polluted days?*

**Response:**

That's a very good comments. Because of the low concentration of isoprene, it was discussed as part of the alkene when comparison was considered in the original manuscript. According to the suggestions of reviewers, we analyzed isoprene separately from other alkenes and found that the emission of isoprene increased significantly during the polluted days, along with the increase of temperature (Figure 3c, 3f). However, concentrations of most aromatics which were regarded as tracers of solvent evaporation remained unchanged (Figure 3c, 3f). Therefore, we think there is no clear evidence that the emissions of solvent evaporation increased on the polluted days.

[Figure]

**Figure 3: Diurnal variations in wind speed (WS), temperature (T), O₃, NOx, and TVOCs on clean and polluted days at the (a) and (b) CB and (d) and (e) DHS sites. Differences in VOC concentrations between clean and polluted days at the (c) CB and (f) DHS sites.**

Note: ALK = alkanes, ALE = alkenes (except isoprene), ISO = isoprene, ALY = alkynes, ARO = aromatics, HALO = Halohydrocarbons, MVK = Methyl Vinyl Ketone, and MACR = Methacrolein.

We have also modified this part in the revised manuscript. Now it reads as follows:

As shown in Figure 3, isoprene concentrations at urban sites increased significantly during the O₃ pollution day, which could due to the stronger plant emission at elevated temperature (Guenther et al., 1993; Guenther et al., 2012; Stavrakou et al., 2014). Concentrations of isoprene oxidation products (i.e., MVK and MACR) as well as most OVOCs also increased in the same period. However, similar concentrations of anthropogenic VOCs are found in clean and polluted days. This indicates a stronger photochemical conversion of VOCs existed in O₃ pollution days, which could due to the more favorable meteorological conditions (i.e., higher temperature and solar radiation).
* * *
*7. Based on Figure 3, the increase in the concentration of OVOC on O3 pollution days is largely driven by the increased concentration of acetone. The authors also show in section 3.2.2 that acetone is mainly from vehicle exhaust and industrial sources. Both primary emission and/or secondary transformation may contribute to the increase of OVOC. Can the authors estimate the contribution from primary emissions?*

**Response:**

We appreciate the reviewer's comments. The sources of OVOCs can be divided into anthropogenic primary sources, anthropogenic secondary sources, biogenic sources and background sources (Li et al., 2014; Wang et al., 2015). The multi-linear regression model was used to analyse the sources of OVOCs in different sites in Xi'an. Ethyne, PAN and isoprene were selected as the tracers of the anthropogenic primary source, the anthropogenic secondary source and the biogenic sources respectively. The equation of the multi-linear regression model is as follows:

$$[OVOCs]=k_0+k_1\times[Ethyne]+k_2\times[PAN]+k_3\times[Isoprene] \qquad (R1)$$

where [Ethyne] represents the concentration of Ethyne, [PAN] represents the concentration of PAN, [Isoprene] represents the concentration of isoprene, $k_0$ represents the background concentration, $k_1$, $k_2$ and $k_3$ are the corresponding coefficients.

Based on the analysis of the multi-linear regression model, we have a deeper understanding of the source of OVOCs during the ozone pollution period. From Figure S1 and S2 we found that the contribution of anthropogenic primary sources to OVOCs on $O_3$ pollution days is more significant.

We have carefully revised this statement in the revised manuscript. Now it reads as follows:

The specialty of OVOCs is that in addition to the primary emissions, OVOCs can also be formed through photochemical oxidation with alkenes and aromatics (Birdsall and Elrod 2011). The sources of OVOCs can be divided into anthropogenic primary sources, anthropogenic secondary sources, biogenic sources and background sources (Li et al., 2014; Wang et al., 2015). Base on the multi-linear regression model results (Figure S1 and S2) we found that the contribution of anthropogenic primary sources to OVOCs on $O_3$ pollution days is more significant.

[Figure]

**Figure S1. Time series of measured OVOCs concentrations and OVOCs calculated from the multi-linear regression model.**

**Note.** The equation of the multi-linear regression model is:

$$[OVOCs]=k_0+k_1\times[Ethyne]+k_2\times[PAN]+k_3\times[Isoprene]$$

where [Ethyne] represents the concentration of Ethyne, [PAN] represents the concentration of PAN,

[Isoprene] represents the concentration of isoprene, $k_0$ represents the background concentration, $k_1$, $k_2$ and $k_3$ are the corresponding coefficients, meas. represents measure.

[Figure]

**Figure S2. Contributions of different sources of OVOCs in different sites in Xi'an base on the multi-linear regression model**.
* * *
*8. Did the NOx concentration change during the polluted and clean days? Did NOx play a role in increasing the O3 concentration on polluted days?*

**Response:**

We appreciate the reviewer's comments. Compared with clean days, the NOx concentration at CB and DHS sites on polluted days increased by 5.8 and 10.3 ppb, respectively. However, before analyzing the influence of NOx changes on ozone, a VOCs-NOx-$O_3$ sensitivity analysis is needed. We have added VOCs-NOx-$O_3$ sensitivity analysis in section 3.4.2. Now it reads as follows:

The relationship between the ozone production rates (P ($O_3$)), anthropogenic VOCs (AVOCs) reactivity and NOx reactivity of the CB, DHS, and QL sites during the observation period was shown in Figure 12. The black curve in the Figure 12 represents the P ($O_3$) contour, and the black straight line represents the connection line of the P ($O_3$) turning point (ridgeline), whose slope represents the photochemical parameter $k_{NOx}/k_{AVOCs}$ (Jiang et al., 2018). When the site's $k_{NOx}/k_{AVOCs}$ value is located above the ridgeline, it means that ozone formation is under VOCs-limited regime, otherwise it means that ozone formation is under NOx-limited regime. It can be seen from Figure 12 that the ozone generation of QL site is located in the NOx-limited regime, and reducing NOx can effectively control ozone generation. The ozone generation of DHS site is located in the VOCs-limited regime, and reducing VOCs can effectively control ozone generation. However, CB site is located in the transition regime between VOC- and NOx-limited regimes. Therefore, simultaneous reduction of VOCs and NOx concentration should be considered at CB site to achieve the purpose of controlling $O_3$.

[Figure]

**Figure 12. The ozone production rate (P ($O_3$)) contours diagram versus anthropogenic VOCs (AVOCs) and NOx using Empirical Kinetic Modelling Approach at CB, DHS, and QL sites.**

We also have added this part of the results in the conclusion section. Now it reads as follows:

The VOCs-NOx-$O_3$ sensitivity analysis results showed that the ozone generation of DHS site is located in the VOCs-limited regime, CB in the transition regime between VOC- and NOx-limited regimes, and QL sites is located in the NOx-limited regime. Therefore, reducing VOCs concentration at DHS site, reducing VOCs and NOx concentration at CB site, and reducing NOx concentration at QL site can effectively control ozone generation.
* * *
*9. Line 184: It is recommended to provide the VOC list and the grid sampling data in the supporting information.*

**Response:**

We appreciate the reviewer's comments. We agree with reviewer and added VOC list and the grid sampling data of the grid sampling sites in the supporting information. Now it reads as follows:

In the VOC grid sampling, 106 VOCs were measured, including 29 alkanes, 11 alkenes, 1 alkyne, 17 aromatics, 35 halohydrocarbons, 12 OVOCs, and carbon disulfide (Table S2-S3).

**Table S2: Measured VOC species in VOC grid sampling sites.**

| Classification | VOC Species | | |
|---|---|---|---|
| | Ethane | 3-Methylpentane | Methylcyclohexane |
| | Propane | n-Hexane | 2,3,4-Trimethylpentane |
| | Iso-butane | 2,4-Dimethylpentane | 2-Methylheptane |
| | n-Butane | Methylcyclopentane | 3-Methylheptane |
| Alkanes | Cyclopentane | 2-Methylhexane | n-Octane |
| | Iso-pentane | Cyclohexane | n-Nonane |
| | n-Pentane | 2,3-Dimethylpentane | n-Decane |
| | 2,2-Dimethylbutane | 3-Methylhexane | n-Undecane |
| | 2,3-Dimethylbutane | 2,2,4-Trimethylpentane | n-Dodecane |

|  |  |  |  |
|---|---|---|---|
|  | 2-Methylpentane | n-Heptane |  |
| Alkenes | Ethene | Cis-butene | Isoprene |
|  | propene | 1,3-Butadiene | cis-2-Pentene |
|  | Trans-2-butene | 1-Pentene | 1-Hexene |
|  | 1-Butene | trans-2-Pentene |  |
| Alkynes | Ethyne |  |  |
| Aromatics | Benzene | iso-Propylbenzene | 1,2,4-Trimethylbenzene |
|  | Toluene | n-Propylbenzene | 1,2,3-Trimethylbenzene |
|  | Ethylbenzene | m-ethyltoluene | m-diethylbenzene |
|  | m/p-Xylene | p-ethyltoluene | p-diethylbenzene |
|  | o-Xylene | 1,3,5-Trimethylbenzene | Naphthalene |
|  | Styrene | o-Ethyltoluene |  |
| Halohydro-carbons | Freon114 | Tetrachloromethane | Bromodicloromethane |
|  | Freon11 | 1,2-Dichloroethane | Benzyl chloride |
|  | Freon113 | Trichloroethylene | 1,2-Dichlorobenzene |
|  | Chloromethane | 1,2-Dichloropropane | trans-1,2-Dichloroethylene |
|  | Vinylchloride | cis-1,3-Dichloropropene | Freon12 |
|  | Bromomethane | trans-1,3-Dichloropropene | Trichloromethane |
|  | Chloroethane | 1,2,4-trichlorobenzene | Dibromo-monochloro-methane |
|  | 1,1-Dichloroethene | 1,1,2-trichloroethane | Tribromomethane |
|  | Dichloromethane | Tetrachloroethene | 1,1,2,2-tetrachloroethane |
|  | 1,1-Dichloroethane | 1,2-Dibromoethane | 1,3-Dichlorobenzene |
|  | cis-1,2-Dichloroethylene | Chlorobenzene | Hexachloro-1,3-butadiene |
|  | 1,1,1-Trichloroethane | 1,4-Dichlorobenzene |  |
| OVOCs | Acrolein | 2-Propanol | Tetrahydrofuran |
|  | Methyl isobutyl ketone | Vinyl Acetate | Methyl methacrylate |
|  | Acetone | Methyl Ethyl Ketone | 1,4-Dioxane |
|  | MTBE | Ethyl Acetate | 2-Hexanone |
| Others | Carbon Disulfide |  |  |

**Table S3: Concentrations of seven VOCs groups in VOC grid sampling sites.**

| site | Alkanes | Alkenes | Alkynes | Aromatics | Halohydrocarbons | OVOCs | Others | TVOCs |
|---|---|---|---|---|---|---|---|---|
| XF | 19.2 | 8.4 | 2.3 | 4.4 | 4.9 | 14.6 | 0.2 | 54.0 |
| CT | 16.1 | 3.6 | 1.8 | 3.3 | 5.1 | 11.5 | 0.1 | 41.4 |
| HC | 15.2 | 2.4 | 1.2 | 2.8 | 4.5 | 12.1 | 0.1 | 38.2 |
| ZYT | 13.7 | 5.0 | 1.8 | 3.0 | 2.0 | 9.7 | 0.1 | 35.4 |
| YT | 8.7 | 3.3 | 1.1 | 1.1 | 3.4 | 16.1 | 0.2 | 33.9 |
| XS | 12.3 | 2.4 | 1.2 | 2.6 | 3.5 | 10.5 | 0.2 | 32.6 |
| GYL | 10.1 | 2.5 | 1.3 | 2.5 | 3.6 | 9.8 | 0.1 | 29.9 |

| | | | | | | | | |
|---|---|---|---|---|---|---|---|---|
| LTC | 10.5 | 4.8 | 2.3 | 2.4 | 2.4 | 6.7 | 0.1 | 29.4 |
| XY | 11.6 | 2.7 | 1.5 | 1.7 | 3.4 | 8.3 | 0.1 | 29.1 |
| JFT | 9.8 | 2.8 | 1.1 | 1.4 | 2.5 | 10.6 | 0.1 | 28.2 |
| RS | 10.6 | 2.4 | 1.4 | 1.3 | 3.5 | 8.6 | 0.1 | 27.9 |
| LT | 8.7 | 3.2 | 0.9 | 0.9 | 3.4 | 10.5 | 0.2 | 27.7 |
| GZ | 10.1 | 2.6 | 1.4 | 1.7 | 3.6 | 7.2 | 0.1 | 26.6 |
| ZZC | 9.8 | 2.0 | 1.2 | 1.2 | 2.5 | 8.3 | 0.1 | 25.0 |
| JDT | 7.5 | 2.9 | 1.1 | 1.3 | 2.9 | 7.0 | 0.1 | 22.8 |
| CAT | 7.3 | 2.3 | 0.8 | 0.9 | 2.7 | 7.9 | 0.1 | 22.2 |
| XX | 8.0 | 1.8 | 1.3 | 0.9 | 2.9 | 5.5 | 0.1 | 20.6 |
| YST | 5.8 | 1.9 | 1.1 | 1.3 | 2.7 | 7.0 | 0.1 | 19.9 |
| WQ | 7.3 | 0.9 | 0.8 | 0.6 | 3.1 | 6.8 | 0.1 | 19.6 |
| XHT | 5.9 | 1.7 | 1.0 | 0.9 | 2.1 | 6.3 | 0.1 | 18.0 |
* * *
*10. Line 190: Did "the overall level" mean "the average concentration"?*

**Response:**

We appreciate the reviewer's comments. We apologize for the vague expression and carefully revised the statement in the revised manuscript. Now it reads as follows:

Compared with the results of the field observation campaign, the VOC concentration at the CB site was closer to the average concentration in Xi'an, and the VOC concentration at the DHS site was significantly higher than the average concentration.
* * *
*11. Lines 192-194: Are there any specific industrial sources near YT? Again, the contribution of primary emissions to OVOCs should be excluded to draw this conclusion.*

**Response:**

We appreciate the reviewer's comments. Based on the analysis of the multi-linear regression model, the anthropogenic secondary sources are more significant for OVOCs in rural sites. The YT site is a rural site and does not have many primary sources of VOCs (Figure R2), so the source of OVOCs is more from aging sources.

[Figure]

**Figure R2. Geographic environment map of TY site.**
* * *
*12. Lines 209-210: These numbers are the slopes of the fitting lines, not the correlation coefficients. Please also revise other places accordingly.*

**Response:**

We appreciate the reviewer's comments. We agree with reviewer and carefully revised the statement according to the reviewer's suggestion. Now it reads as follows:

The ratios of toluene to benzene at the CB, DHS, QL, and gridded sampling sites were 0.7 ($R_{Pearson}$=0.5), 2.3 ($R_{Pearson}$=0.6), 0.5 ($R_{Pearson}$=0.6), and 1.2 ($R_{Pearson}$=0.9), respectively.
* * *
*13. Lines 215-217: For the grid sampling, only samples at 7:00 and 15:00 were collected. It is reasonable that vehicle exhaust greatly contributed to the overall VOCs because of the sampling time. The authors should state the weakness of the sampling as a caveat.*

**Response:**

We appreciate the reviewer's comments. We agree with reviewer that the sample time may affect the ratio of T/B. In order to reduce the influence of sample time and photochemical reaction on the ratio of benzene to toluene, this study selected the weaker photochemical reaction period (3:00-7:00) for the analysis of toluene and benzene (Figure 5).

[Figure]

**Figure 5: Linear correlations between toluene and benzene at the CB, DHS, QL, and gridded sampling sites between 3:00-7:00 during the observation period.**

We agree with reviewer and carefully revised the statement in the revised manuscript. Now it reads as follows:

In order to reduce the influence of photochemical reaction on the ratio of benzene to toluene, this study selected the weaker photochemical reaction period (3:00-7:00) for the analysis of toluene and benzene (Figure 5). Figure 5 shows that the ratios of toluene to benzene at the CB, DHS, QL, and gridded sampling sites were 1.1 ($R_{Pearson}$=0.5), 3.6 ($R_{Pearson}$=0.6), 0.5 ($R_{Pearson}$=0.8), and 1.75 ($R_{Pearson}$=0.9), respectively. In the urban areas (CB and DHS sites), most of the T/B ratios were distributed within the reference range of vehicle emissions and industrial emissions (Figure 5a, 5b), implying that vehicle sources and industrial sources contribute significantly to the VOCs in Xi'an urban area. In addition, the T/B value of some samples is greater than 5.8 in urban area which may affected by paint solvent usage source (Figure 5b). However, the detailed source contribution needs to be obtained through PMF source analysis results (Section 3.2.2). In the rural area (QL site), most of the T/B ratios were distributed within the reference range of vehicle emissions and burning emissions (Figure 5c), implying that vehicle sources and burning sources contribute significantly to the VOCs in Xi'an rural area. In the gridded sampling sites, the T/B ratio was predominately concentrated around 1.5, indicating that vehicle exhaust sources may greatly contributed to the overall VOCs in Xi'an (Figure 5d).
* * *
*14. MVK is a photochemical product of isoprene. Why are most of the MVK attributed to vehicle exhaust at the CB site?*

**Response:**

We appreciate the reviewer's comments. As the photooxidation product of isoprene, MVK could be emitted from secondary sources. However, the tunnel and vehicular exhaust emissions measurements have indicated that MVK could be emitted from vehicle exhaust sources including gasoline and diesel vehicles (Biesenthal and Shepson, 1997;Kean et al., 2001;Ling et al., 2019). In urban site with less affected by biological sources, MVK may more affected by anthropogenic emissions
* * *
*15. Figure 1: it is recommended to mark Feiwei plain.*

**Response:**

We appreciate the reviewer's comments. We agree with reviewer and marked the Feiwei plain in Figure1. Now it reads as follows:

[Figure]

**Figure 1: VOC field observation and grid sampling sites in Xi'an.**

Note: The topographic image was obtained from Google Earth.
* * *
*16. Figures 5 and 7: Linear correlations are shown, not correlation coefficients.*

**Response:**

We appreciate the reviewer's comments. We agree with reviewer and carefully revised the statement according to the reviewer's suggestion. Now it reads as follows:

**Figure 5: Linear correlations between toluene and benzene at the CB, DHS, QL, and gridded sampling sites.**

**Figure 7: Linear correlations between (a) iso-pentane and n-pentane and (b) propane and ethane at the CB (red), DHS (orange), QL (blue), and gridded sampling sites (light blue).**
* * *
*17. Figure 6: What does the green line represent?*

**Response:**

We appreciate the reviewer's comments. The green line represents the initial emission ratio of m/p-xylene and ethylbenzene, which can be replaced by the highest concentration ratio in periods where the photochemical reaction is weak.

We have added the green line description in the caption in the revised manuscript. Now it reads as follows:

[Figure]

**Figure 6: Diurnal variations in m/p-xylene to ethylbenzene and OH exposure at the CB, DHS, QL, and gridded sampling sites.**

Note: Time is expressed in CST. The green line represents the initial emission ratio of m/p-xylene and ethylbenzene.
* * *
*18. Figure 8: What do the bars and dots represent? Please explain in the caption.*

**Response:**

We appreciate the reviewer's comments. We agree with reviewer and add the explanation in the caption.

Now it reads as follows:

**Figure 8: Source profiles and contributions of VOCs in the CB, DHS, and QL sites during the observation period.**

Note. Bars represent the concentration of each species apportioned to the factor, and black dots represent the percent of each species apportioned to the factor.
* * *
***Technical comments***

*1. Line 10: "a critical precursors"… delete "a".*

**Response:**

We appreciate the reviewer's comments. We are sorry for our mistakes, and we have removed "a" in this sentence.
* * *
*2. Line 23: References are missing.*

**Response:**

We appreciate the reviewer's comments, and we have added references in this sentence in the revised manuscript. Now it reads as follows:

Atmospheric pollution in China is characterized by frequent secondary pollution, which is primarily reflected by the yearly increasing ozone ($O_3$) concentrations and proportion of secondary organic components (SOA) in $PM_{2.5}$ (Lu et al., 2018; Huang et al., 2014).
* * *
*3. Line 35: "source" should be "sources"*

**Response:**

We appreciate the reviewer's comments. We are sorry for our mistakes, and we have change *"source" to "sources"* in this sentence.
* * *
*4. Line 36: "indicates" should be "indicate"*

**Response:**

We appreciate the reviewer's comments. We are sorry for our mistakes, and we have change *"indicates" to "indicate"* in this sentence.
* * *
*5. Equation 1, lines 98 and 100: the rate coefficient is typically represented by the lowercase k.*

**Response:**

We appreciate the reviewer's comments. We are sorry for our mistakes, and we have change the rate coefficient K by the lowercase k in the revised manuscript.
* * *
**Lastly, we would again express our appreciation to the reviewers and editor for their warm-hearted help. Thank you very much!!!!**
* * *

---

## Author Response (AR2)

**Response to the Comments of the Reviewers**

Dear Editor and Reviewers,

We would like to thank you and the reviewers for the great efforts and elaborate work on this manuscript.

We revised the manuscript by responding to each of the suggestions in the reviews. In our response, the questions of the reviewers are shown in *Italic* form and the responses in standard form.

We appreciate your help and time.

Sincerely yours,

Xin Li and Co-authors.

College of Environmental Sciences and Engineering
Peking University
100871 Beijing China
E-mail: li_xin@pku.edu.cn
Tel: +86-185 1358 6831
* * *
*Manuscript Number: acp-2020-704.*

*Manuscript Title: Spatiotemporal Variation, Sources, and Secondary Transformation Potential of VOCs in Xi'an, China.*
* * *
**Response to Reviewer #1**

***General comments***

*1. I do appreciate the time and effort that the authors put into revising the manuscript and replying to my comments. The authors have sufficiently addressed most of my comments. In particular, the additional information presented in Section 3.2.1 and Section 3.4.2 is very helpful.*

*I only have some concerns about the multi-linear regression model that the authors used to investigate the source contributions of OVOCs. Overall, the results are appealing. However, the authors lumped all the anthropogenic primary sources into one factor, and all the anthropogenic secondary sources into another factor. I wonder whether the chemical species used for different factors are representative. Is ethyne a good tracer for all the primary sources, at least, for the major primary sources? And can PAN represent the major secondary sources? Have the authors tried to replace ethyne and PAN with other "representative" tracers to see if the results are similar?*

**Response:**

We appreciate the reviewer's comments. This is a good point. In the multi-linear regression analysis, we have also tried other tracers to explore the source of OVOCs. In previous studies, ethyne (Liu et al., 2009; Yuan et al., 2010; Yuan et al., 2012), CO (Rappenglueck et al., 2010), and benzene (Zhu et al., 2019; Xia et al., 2021) was often used as a tracer for anthropogenic primary sources in the atmosphere. Moreover, PAN (Rappenglueck et al., 2010; Li et al., 2014; Xu et al., 2020) and $O_3$ (Li et al., 2014) was often used as a tracer for anthropogenic secondary sources in the atmosphere. We cross-combined these typical tracers and obtained six multi-linear regression model equations (R1)-(R6) as follows:

Type 1: $[OVOCs]=k_0+k_1\times[Ethyne]+k_2\times[PAN]+k_3\times[Isoprene]$         (R1)

Type 2: $[OVOCs]=k_0+k_1\times[CO]+k_2\times[PAN]+k_3\times[Isoprene]$         (R2)

Type 3: $[OVOCs]=k_0+k_1\times[Benzene]+k_2\times[PAN]+k_3\times[Isoprene]$      (R3)

Type 4: $[OVOCs]=k_0+k_1\times[Ethyne]+k_2\times[O_3]+k_3\times[Isoprene]$       (R4)

Type 5: $[OVOCs]=k_0+k_1\times[CO]+k_2\times[O_3]+k_3\times[Isoprene]$        (R5)

Type 6: $[OVOCs]=k_0+k_1\times[Benzene]+k_2\times[O_3]+k_3\times[Isoprene]$      (R6)

From the analysis results of different types of multi-linear regression model, anthropogenic primary sources were the primary source of OVOCs concentration at CB and DHS sites (Figure R1). Then, the contribution of anthropogenic secondary sources

and background sources to OVOCs is more significant at QL sites (Figure R1). Although the proportion of OVOC sources will be slightly different of different types of multi-linear regression fitting (Figure R1), they are all within the model error range (Table R1).

[Figure]

**Figure R1. Contributions of different sources of OVOCs in different sites in Xi'an base on the different types of multi-linear regression model.**

**Note.** T1 to T6 represent Type 1 (R1) to Type6 (R6), respectively.

**Table R1. Contributions of different sources of OVOCs at CB, DHS and QL sites during the observation period base on the different types of multi-linear regression model.**

| Sites | Background (%) | Anthropogenic primary (%) | Anthropogenic secondary (%) | Biogenic (%) | $R_{pearson}$ | Type |
|-------|----------------|---------------------------|------------------------------|--------------|---------------|------|
|       | 25.9±10.7 | 50.4±19.3 | 18.8±18.2 | 4.9±11.2 | 0.72 | Type 1 |
|       | 24.1±8.4 | 63.6±13.5 | 9.2±11.9 | 3.1±8.2 | 0.56 | Type 2 |
| CB    | 28±11.6 | 47.3±17.9 | 19.5±17.1 | 5.2±12 | 0.55 | Type 3 |
|       | 25.2±8.4 | 46.2±21.8 | 27.6±19.5 | 1±2.6 | 0.57 | Type 4 |
|       | 24.9±7.6 | 56.4±19.7 | 16.9±16.5 | 1.8±4.8 | 0.53 | Type 5 |
|       | 22.8±8 | 44.1±22 | 31.4±19.5 | 1.7±4.8 | 0.58 | Type 6 |
|       | 24.9±17 | 41.5±21.3 | 24.9±20.2 | 8.7±10.4 | 0.83 | Type 1 |
| DHS   | 21.3±6.7 | 54.8±15.7 | 15.5±15.4 | 8.4±10.9 | 0.55 | Type 2 |
|       | 22.4±9.7 | 46±20.4 | 23.7±18.2 | 8±9.3 | 0.78 | Type 3 |

| | | | | | | |
|---|---|---|---|---|---|---|
| | 22.8±7 | 39.7±19.5 | 30.4±18.3 | 7.1±9.5 | 0.83 | Type 4 |
| | 21.1±5.9 | 51.1±19.1 | 20.2±17.4 | 7.6±10 | 0.79 | Type 5 |
| | 20.9±8.4 | 44.4±19.6 | 28.7±17.6 | 6±7.9 | 0.73 | Type 6 |
| | 39.1±16.7 | 17.3±18.7 | 35.6±24.3 | 8±12.3 | 0.89 | Type 1 |
| | 39.7±16.2 | 20.6±20.6 | 34.2±24.5 | 5.5±9.1 | 0.78 | Type 2 |
| | 43.3±19.6 | 15±17.1 | 34.5±23.5 | 7.2±11.4 | 0.79 | Type 3 |
| QL | 35.5±11.8 | 19±19.8 | 38.6±23.3 | 7±11.5 | 0.67 | Type 4 |
| | 31.7±9.7 | 29.4±28.7 | 34±27.8 | 4.9±8.7 | 0.59 | Type 5 |
| | 38.1±13.9 | 19.5±20.5 | 37.9±22.1 | 4.5±8.3 | 0.64 | Type 6 |

Ethyne and CO are mainly come from vehicle exhaust sources, combustion sources and industrial combustion sources (Ling et al., 2011; Liu et al., 2008). Benzene mainly comes from vehicle exhaust sources, industrial sources, and paint solvent usage sources (Liu et al., 2008; Zhu et al., 2019; Xia et al., 2021). When analysing the source of OVOCs in the Pearl River Delta, Zhu et al. (2019) and Xia et al. (2021) used benzene as multi-linear regression model tracers for anthropogenic primary sources. This is mainly due to the relatively large contribution of solvent sources and industrial sources to OVOCs in the Pearl River Delta region (Ling et al., 2011). Base on PMF model results, anthropogenic emissions of VOCs in Xi'an are rather complicated, with vehicle exhaust sources, industrial sources, and combustion sources contributing significantly to VOCs. Moreover, when ethyne is used as the tracer of anthropogenic primary sources for fitting, the fitting effect ($R_{pearson}$) of multi-linear regression model is significantly better than that of CO and benzene (Table R1). Compared with CO and benzene, ethyne can better reflect the anthropogenic primary emissions of OVOCs in this study. In addition, $O_3$ has a definite regional background which is very difficult to quantify (Li et al., 2014). Compared to $O_3$, the background concentrations of PAN are negligible which makes PAN a better tracer for anthropogenic secondary sources (Rappenglueck et al., 2010; Li et al., 2014; Xu et al., 2020).

Based on the above discussion, ethyne and PAN were selected as multi-linear regression model tracers for anthropogenic primary sources and anthropogenic secondary sources in this study, respectively.

*2. In Figure S2, there are three columns for CB and DHS. What do those three columns represent?*

**Response:**
As we described in the manuscript (Line 208-Line 209), there were two $O_3$ pollution events in the Xi'an urban area (CB and DHS sites), from June 23 to 26, 2019 and from June 30 to July 15, 2019. Therefore, we define the period of this ozone pollution event as a polluted condition, and the rest of the period as a clean condition. **"Total"** represent the entire observation period. We have also revised the caption of Figure S2 to make it clearer. Now it reads as follows:

**Figure S2. Contributions of different sources of OVOCs in different sites in Xi'an base on the multi-linear regression model**. "Total" represent the entire observation period. The "Polluted" represent the period of ozone pollution event at CB and DHS sites (from June 23 to 26, 2019 and from June 30 to July 15, 2019), and the rest of the period as a "Clean" condition.
* * *
*Minor comments*

*1. Line 77 in the revised manuscript regarding GC methods: It should be "later", instead of "latter".*

**Response:**

Revised accordingly.
* * *
*2. Figure S2: "total", instead of "totol"?*

**Response:**

Revised accordingly.
* * *

[revised manuscript text omitted]